

# Accounting for rain type non-stationarity in sub-daily stochastic weather generators

Lionel Benoit[1], Mathieu Vrac[2], and Gregoire Mariethoz[1]

[1]Institute of Earth Surface Dynamics (IDYST), University of Lausanne, Lausanne, Switzerland
[2]Laboratory for Sciences of Climate and Environment (LSCE-IPSL), CNRS/CEA/UVSQ, Orme des Merisiers, France

**Correspondence:** Lionel Benoit (lionel.benoit@unil.ch)

**Abstract.** At sub-daily resolution, rain intensity exhibits a strong variability in space and time, which is favorably modelled using stochastic approaches. This strong variability is further enhanced because of the diversity of processes that produce rain (e.g. frontal storms, mesoscale convective systems, local convection), which results in a multiplicity of space-time patterns embedded into rain fields, and in turn leads to non-stationarity of rain statistics. To account for this non-stationarity in the
context of stochastic weather generators, and therefore preserve the climatological coherence of rain simulations, we propose to resort to rain types simulation.

We explore two methods to simulate rain type time series conditional to meteorological covariates: a parametric approach based on a non-homogeneous semi-Markov chain, and a non-parametric approach based on multiple-point statistics. Both methods are tested by cross-validation using a 17-year long rain type time series defined over central Germany. Evaluation results
indicate that the non-parametric approach better simulates the relationships between rain types and meteorological covariates. Indeed, the inherent simplifications in the parametric model do not allow fully resolving complex and non-linear interactions between the rainfall statistics and meteorological covariates.

The proposed approach is applied to generate rain type time series conditional to meteorological covariates simulated by a Regional Climate Model under an RCP8.5 emission scenario. Results indicate that, by the end of the century, the distribution
of rain types could be modified over the area of interest, with an increased frequency of convective- and frontal-like rains at the expense of more stratiform events.

## 1    Introduction

Stochastic weather generators are statistical models designed to simulate realistic random sequences of atmospheric variables (e.g. temperature, rain and wind). Their main target is to reproduce both the internal variability of each variable of interest, and
the relationships between these variables (Wilks and Wilby, 1999; Furrer and Katz, 2007; Ailliot et al., 2015). These features make stochastic weather generators particularly well suited for producing synthetic climate histories in view of impact studies (Mavromatis and Hansen, 2001; Verdin et al., 2015; Paschalis et al., 2014), as well as for stochastic downscaling of climate projections (Burton et al., 2010; Wilks, 2010; Volosciuk et al., 2017). Within stochastic weather generators, rainfall has long been recognized as a critical variable, in particular because of the strong intermittency (Pardo-Igúzquiza et al., 2006; Schleiss





et al., 2011) and variability (Smith et al., 2009; Gires et al., 2014) of the rain process. The apparent intermittency and variability of rainfall increase with the time resolution of interest (Krajewski et al, 2003; Mascaro et al., 2013), and if resolutions of the order of 1 h (or finer) are considered, it appears that storms caused by different generation processes (e.g. frontal storms, mesoscale convective systems, local convection) result in different rain field organizations and temporal patterns (Emmanuel

et al., 2012; Marra and Morin, 2018).

Such changes in rainfall characteristics make rain statistics time-varying. In terms of stochastic modelling, this implies that the stochastic process used to model rainfall is non-stationary through time, i.e. the parameters of the stochastic model change over time. The most common way to deal with the non-stationarity of rain statistics is to define a priori (i.e. prior to model calibration) the time periods during which stationarity is assumed. Afterwards, a piecewise-stationary modelling is applied, i.e.

model parameters are kept constant within a single stationary period, but are allowed to vary between stationary periods. The temporal scale at which non-stationarity occurs is defined by the modeler according to prior knowledge and assumptions about the rain process at hand, and ranges from seasons (Paschalis et al., 2013; Bárdossy and Pegram, 2016; Peleg et al., 2017) to single rain storms (Caseri et al., 2016; Benoit et al., 2018a).

However, several empirical studies have shown that at the sub-daily scale, rain statistics can change at a higher rate than

alleged in most piecewise-stationary stochastic rainfall models. More precisely, rain statistics have been shown to abruptly change within a single day (Emmanuel et al., 2012), and even within a single rain storm (Kumar et al, 2011; Ghada et al., 2019). To model rain non-stationarity on a more data-driven basis, and thereby account for the sub-daily non-stationarities reported above, it has recently been proposed to classify rain fields into rain types (e.g. based on weather radar images) prior to stochastic modelling (Lagrange et al., 2018; Benoit et al., 2018b). Rain fields belonging to the same rain type are then

deemed statistically similar, and periods with a constant rain type can be regarded as stationary periods for the simulation of rain intensity.

The main goal of this paper is to propose a framework that extends the use of rain types to encode rain non-stationarity in the context of stochastic weather generation. More precisely, it develops a method for stochastic simulation of rain type time series conditional to the state of the atmosphere, i.e. conditional to meteorological variables such as pressure, temperature, humidity

or wind. The advantage of such an approach is twofold: firstly, using a stochastic simulation to generate rain types allows to properly reproduce the natural variability of rain type occurrence, and thereby to indirectly model the non-stationarity of rain statistics observed in historical datasets. Secondly, the conditioning of the stochastic rain type model to the state of the atmosphere preserves the relationships between rain type occurrence and the value of meteorological covariates, which ensures the climatological coherence of the stochastic weather generator. Once realistic rain type time series have been simulated,

high-resolution rain fields can be simulated conditional to rain types using any high-resolution stochastic rainfall generator (Leblois and Creutin, 2013; Paschalis et al., 2013; Nerini et al., 2017; Benoit et al., 2018a).

The remainder of the paper is structured as follows. First, Sect. 2 presents an example of sub-daily rain type time series, and Sect. 3 proposes two different stochastic models to capture the main statistical features of this dataset. Next, Sect. 4 compares the performance of these two models through a cross-validation procedure, and applies the most reliable model to





the downscaling of EURO-CORDEX RCM precipitation future projections as an illustration of the method. Finally, Sect. 5 provides some conclusions about stochastic rain type modelling.

## 2    Example dataset of rain type time series

Before proposing stochastic models able to mimic the rain type occurrence process (Sect. 3), the present section explains how
rain type time series are derived from weather radar observations, and investigates the main features of rain type occurrence in a mid-latitude climate.

We focus hereafter on a 100 km x 100 km squared area centered on the city of Jena in the Land of Thüringen, Germany (Fig. 1a). This area has been chosen because its flat topography and its location far from coastlines or major topographic barriers ensure spatially homogeneous rain fields, allowing to focus on the temporal component of rainfall non-stationarity.
Over this area, data used for rain typing consists of radar images extracted from the RADOLAN (RAdar-OnLine-ANeichung) dataset (Winterrath et al., 2012; Kaspar et al., 2013), which is provided in open access by the German meteorological agency (Deutscher Wetterdienst - DWD). It consists of raw (i.e. not adjusted on rain gauges) radar image composites over entire Germany from 01/01/2001 to present. RADOLAN radar image resolution is 1 km x 1km in space, and 5 min in time. In practice, however, we resampled radar images at a 10 min resolution and restricted our study to the period 01/01/2001 –
15   31/12/2017.

The RADOLAN dataset is used as baseline information for rain typing, following the approach proposed by Benoit et al. (2018b). This rain typing method consists of classifying radar images according to their space-time statistical signature. To this end, rainy time steps are first defined as periods with more than 10% radar pixels measuring rain. The other time steps are classified as dry, and are not considered for rain typing. Next, 10 statistical metrics are computed for each rainy radar image
in order to assess the space-time-intensity behavior of the rain field observed in the image. Among these metrics, 3 relate to the statistical distribution of rain intensity observed in the radar image, 3 characterize the spatial arrangement of rain patterns within the image, and 4 evaluate the temporal evolution of the rain field due to rain advection and diffusion between consecutive periods. Then, the 10 metrics are used as a basis for classification using a Gaussian Mixture Model (GMM). All details about the 10 metrics and the clustering approach with GMM can be found in Benoit et al. (2018b). The resulting clusters correspond
to rain fields with similar space-time behaviors. The number of rain types is selected as a compromise between goodness of fit to rain field statistics and model parsimony. In the present case, the parsimony is favored in order to allow for a physical interpretation of the resulting rain types. As a result, 6 rain types are identified in the example dataset (Fig. 1b). Among them, rain types 1 and 4 correspond to rather stratiform and spread rain events, rain type 3 corresponds to frontal rain storms, and rain types 5 and 6 can be associated with rather convective rains. Rain type 2 cannot be associated to a specific rain behavior,
but rather gather rain fields that are not classified otherwise, and often correspond to partial rain coverage.

Fig. 2 investigates the occurrence of rain types through time, and highlights some features. Fig. 2a displays the frequency of each rain type at the seasonal scale. The non-rain type (T0 in Fig. 2c) is not explicitly shown here in order to focus on rainfall, but it should be noticed that dry periods (i.e. less than 10% rain coverage in the radar image) represent between 89% and 97%



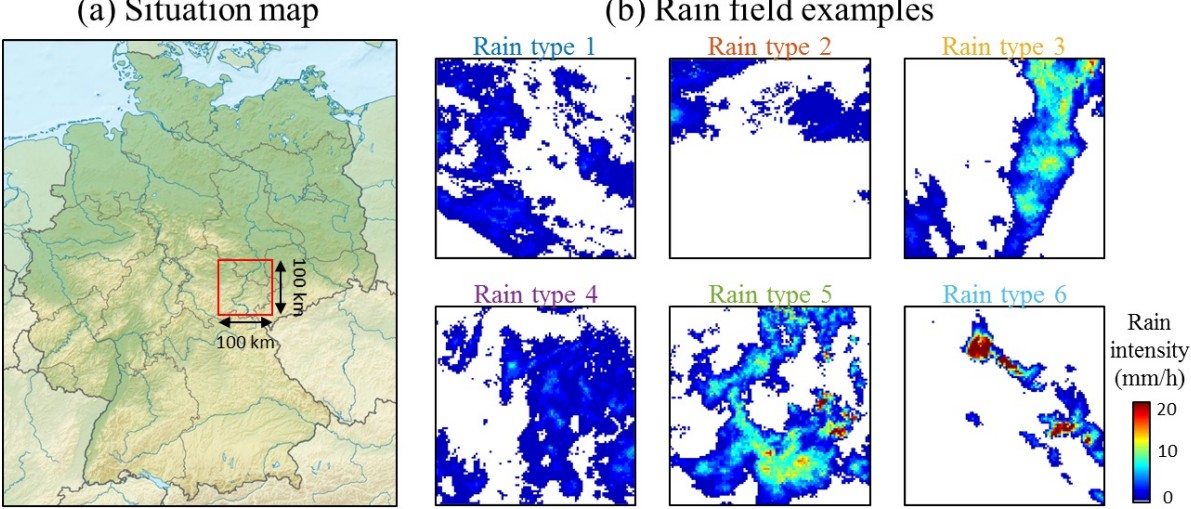

**Figure 1.** Radar dataset used for rain typing. (a) Study area. The red square denotes the area of interest, centered on Jena, Thüringen, Germany. (Background map from Wikipedia.org, licensed under CC BY 3.0) (b) Example of RADOLAN radar images (cropped over the area of interest) for each rain type.

of each season. It appears from Fig. 2a that the frequency of occurrence of individual rain types is strongly variable across the year. For example, stratiform rain types 1 and 4 occur mostly in winter, while convective rain types 5 and 6 are most common in summer. In addition, one can notice a strong inter-annual variability in rain occurrence, summers 2003 and 2011 having a low occurrence of rain, while rain occurrence is particularly high during winters 2006 and 2011. Figure 2b displays the

Cumulative Density Function (CDF) of the duration of each rain type. Each curve in Fig. 2b corresponds to the probability that a rain event of a given type does not exceed the duration given in abscissa. This figure shows that all rain types are persistent in time with durations ranging from few minutes to more than 3 hours, and that some types (e.g rain types 4 and 5) are more persistent than others (e.g. rain types 2 and 3). Finally, Fig. 2c displays the empirical transition matrix between rain types, and focuses on inter-type transitions (i.e. transitions to the same type are ignored and denoted by red crosses). This figure shows

that the patterns of transition between rain types are complex, and that the transitions involving type 0 (i.e. no rain) are largely dominant.

    The strong seasonality and inter-annual variability of rain type occurrence emerging from Fig. 2a can be explained to a large extent by regional meteorological conditions (Vrac et al., 2007; Willems, 2001; Rust et al., 2013). Hereafter, we investigate the links between rain type occurrence and a set of 7 meteorological covariates that are deemed to influence rainfall behavior,

namely: (1) mean daily sea level pressure, (2-4) mean, minimum and maximum daily temperature at 2 m, (5) mean daily relative humidity, and (6-7) mean daily Eastward and Northward components of synoptic wind at 850 hPa. The actual values of the meteorological covariates used in this study are extracted from the ERA-5 reanalyzes (Hersbach et al., 2018). To be compatible with the temporal resolution of RCM projections used for the illustration of our framework (see Sect. 4.2), only daily values are



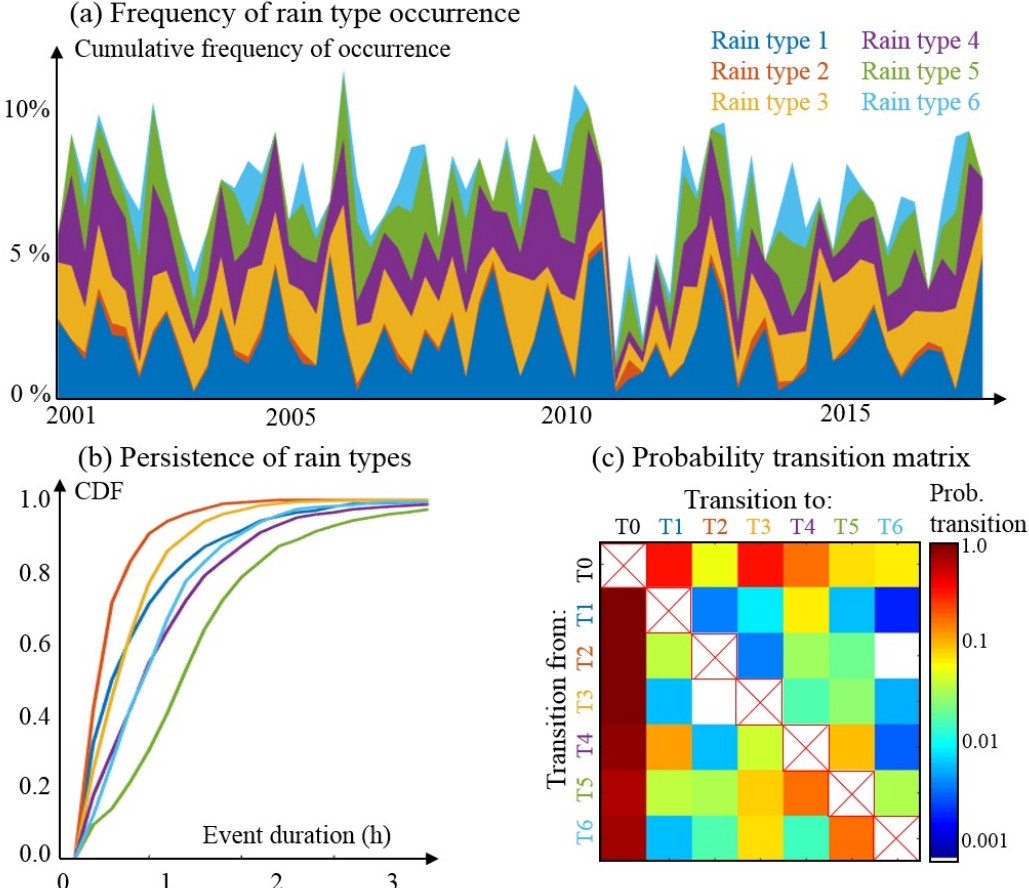

**Figure 2.** Main features of a rain type time series (2000-2017) observed over central Germany. (a) Frequency of rain type occurrence computed at a seasonal basis. (b) CDF of event duration stratified by rain type. (c) Empirical matrix of transition probability between rain types.

extracted from the ERA-5 dataset. They are disaggregated afterwards to the resolution of rain type data (i.e. 10 min). To this end, the mean daily values are assumed to occur at 12 PM local time, and are then interpolated at a 10 min resolution using a polynomial interpolation. Daily temperature variations are represented by a spline that reaches the minimum daily temperature at 5 AM local time and the maximum daily temperature at 3 PM local time. Note that all the above-mentioned meteorological data are averaged over the whole area of interest before further use. Figure 3 displays the influence of meteorological covariates on rain type occurrence, and confirms the strong dependence of rain type occurrence on temperature and wind speed, and to a lower extent on pressure, relative humidity and wind direction.





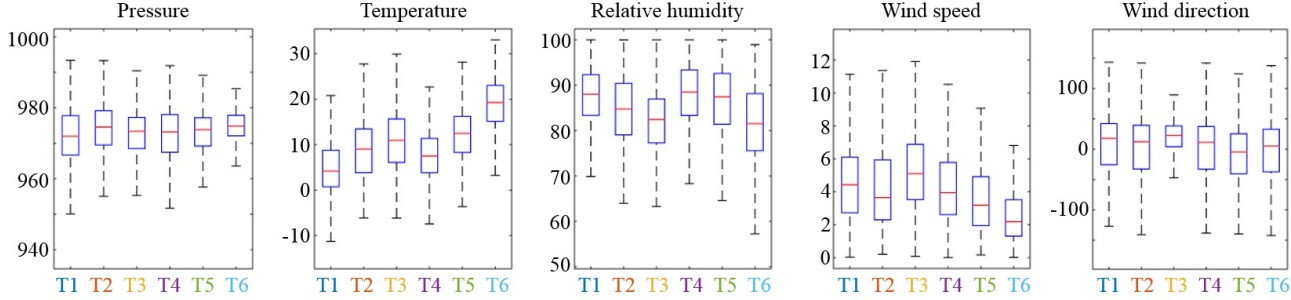

**Figure 3.** Statistics of meteorological covariates for each rain type. The meteorological data have been extracted from the ERA-5 reanalysis dataset.

## 3 Stochastic rain type models

### 3.1 Building a Markov-chain-based model of rain type occurrence

The first approach we consider to model rain type occurrence builds on Markov-chain models, which were originally developed to model dry/wet sequences in daily resolution applications (Richardson, 1981; Wilby, 1994; Wilks and Wilby, 1999). The
existing models are substantially amended to match with the features of sub-daily resolution rain type time series, which significantly differ from dry/wet sequences at daily resolution. To model sub-daily resolution rain type time series, we adopt a non-homogeneous semi-Markov model (i.e. with non-stationary transition probabilities and time-varying step lengths) with N+2 states (Fig. 4). Among these N+2 states, N states model rain types (in the example dataset above N=6), and 2 states model dry periods that are split into 'short dry' (duration <24h) and 'long dry' (duration >24h) states. The 'long dry' state can only
transition to 'short dry', and the rain types (as well as the 'short dry' type) can transition to each other and with the 'short dry' type. A semi-Markov approach (Foufoula-Georgiou and Lettenmaier, 1987; Bárdossy and Plate, 1991) is used to account for the persistence of rain (and 'short dry') types. In our model, the duration of these types is explicitly defined by a Probability Density Function (PDF) of event duration, and the Markov chain is not allowed to be twice in the same state (i.e. transitions from state i to i are censored). On the contrary, the 'long dry' state always lasts exactly 24h, but is allowed to transition to
itself to generate long lasting dry spells. Finally, to account for the non-stationarity of rain type occurrence in time, the semi-Markov chain is made non-homogeneous (Hughes and Guttorp, 1999; Vrac et al., 2007). It consists of changing the probability transition matrix of the Markov chain over time conditionally to a set of meteorological covariates X:

$$P\left(S_t | S_{t-1} = i, X_t\right) \propto \gamma_{ij}.\exp\left(-\frac{1}{2}(X_t - \mu_{ij})\Sigma^{-1}(X_t - \mu_{ij})^T\right) \tag{1}$$

Where $S_t$ is the state of the Markov chain at time $t$, $\Sigma$ is the covariance matrix of the covariates, $\mu_{ij}$ is the mean vector
of the covariates when the transition from type i to type j occurs, and $\gamma_{ij}$ is the baseline (i.e. long term averaged) transition probability from state i to state j. It should be emphasized here that since the 'long dry' state is allowed to transition to itself, the





probability of transition from 'long dry' to 'long dry' is driven by the meteorological covariates, and indirectly the length of dry spell duration is made dependent on the state of the atmosphere. Conversely, the persistence of all other states (i.e. rain types and 'short dry' type) is stationary in time, and only the probabilities of occurrence of these states depend on meteorological conditions. The non-homogeneous semi-Markov model of rain type occurrence is summarized in Fig. 4.

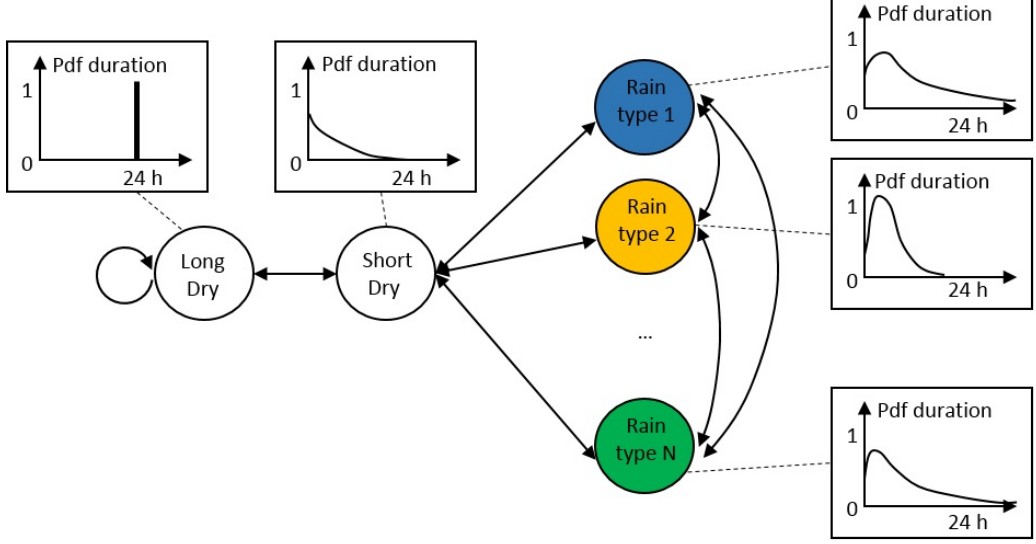

**Figure 4.** Schematic view of the non-homogeneous semi-Markov model used to model sub-daily rain type occurrence.

The parameters of the model can be inferred from historical records using the following procedure. First, the baseline transition matrix is estimated by counting all transitions between each pair of rain types (incl. dry states) occurring in the calibration dataset, and by normalizing the result by the total number of transitions. Then, the parameters required to make the transition matrix non-homogeneous (i.e. $\mu_{ij}$ and $\Sigma$, cf Eq. (1)) are estimated using a synchronous record of covariate observations. More precisely, $\Sigma$ is obtained by computing the empirical covariance matrix of the covariates for the whole
calibration dataset, while the $\mu_{ij}$ coefficients are estimated by computing the mean value of the covariates for the time steps where the transition from i to j occurs. Finally, the PDFs of rain type duration are assumed to be gamma distributions whose parameters are inferred by likelihood maximization using the observed rain type durations.

After inference of model parameters, the calibrated model can be used to generate synthetic rain type time series conditioned to meteorological covariates. To this end, time series of covariates have to be available for the target simulation period. Next,
synthetic rain types are stochastically generated by iteratively (i) simulating a rain type (or dry / wet) transition using Eq. (1), and (ii) generating the duration of the current event by sampling the PDF of duration of the rain (or dry) type of interest.





## 3.2 Non-parametric resampling using Multiple-Point Simulations

Because sub-daily rain type time series and their dependence to atmospheric conditions are complex, one can suspect that some part of their behavior cannot be fully captured by a parametric model (Oriani et al., 2018) as the one described above, even if the model is relatively sophisticated. One option to account for higher-order properties when generating synthetic rain type

time series is to adopt a non-parametric approach based on the resampling of historical datasets.

To do so, we adopt the framework of multiple-point simulations (MPS). MPS consist of using a training dataset (here a past rain type record) to estimate empirically the probability distribution of the variable of interest (here rain type occurrence at a given time step) conditionally to the values already simulated in its temporal neighborhood (Fig. 5a). In the specific MPS algorithm we use (Gravey and Mariethoz, 2018, Under Review), the conditional pdf is indirectly assessed by making a random

sampling of the training dataset that aims at finding a pattern that is similar to the local conditioning neighborhood (Fig. 5b). In practice, we use a 100 h neighborhood for the present application. Once a match is found (i.e. a pattern in the training image that minimizes the Hamming distance with the target pattern), the corresponding value is imported in the simulation grid (Fig. 5c), and the procedure is iterated until the full simulation grid is filled. In the MPS framework, the dependence of rain type occurrence to meteorological covariates can be handled by multivariate-MPS simulation (Mariethoz et al., 2010). It

consists of stacking time series of meteorological covariates with the time series of rain type occurrence, and to evaluate the conditioning neighborhood on the time series of the resulting vector-variable. Here we use a simplified version of multivariate-MPS where only the co-located covariates (i.e. the values of the covariates observed at the exact time step to simulate) are accounted for during the matching procedure.

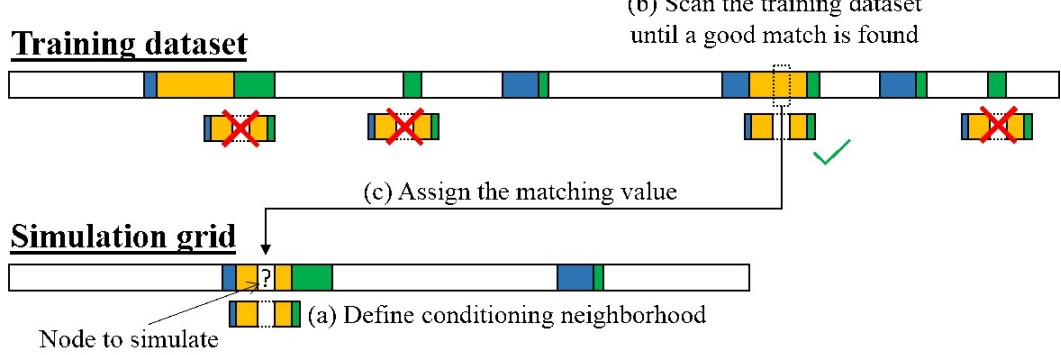

**Figure 5.** Schematic view of the MPS algorithm used for non-parametric resampling of historical rain type time series.

Since MPS is a non-parametric approach, it does not require model calibration strictly speaking. Instead, it requires a training

dataset to resample, which should include both the variable of interest (here a rain type time series) and optional covariates (here meteorological covariates). To produce reliable results, and in particular meaningful uncertainty estimates, MPS requires large training datasets (Emery and Lantuéjoul, 2014). In the present case, the training dataset is the historical record of joint rain types and meteorological covariates observations available over the target area (17 years). After selection of a training





dataset, simulations are obtained by resampling the training dataset using the MPS algorithm described above. As is the case for the parametric model, a time series of meteorological covariates has to be available for the target simulation period in order to condition the simulated rain type time series to the state of the atmosphere.

## 4 Results

### 4.1 Assessment of the stochastic rain type models by cross-validation

The performance of both stochastic rain type models is assessed by cross-validation, using the dataset introduced in Sect. 2. In practice, we adopt a leave-one-year-out procedure. For a given simulation year, the rain type model is first calibrated using data from the 2001-2017 period, excluding the year to simulate. Next, 50 realizations of rain type time series are generated for the year of interest using the previously calibrated stochastic model, conditioned to observations of the meteorological covariates derived from the ERA5 reanalysis as described in Sect. 2. Finally, the same procedure is iterated for each year of the test period (i.e. 2001-2017), and 50 realizations of 17-years-long rain type time series are obtained by concatenating in time the 17 yearly simulations.

For model evaluation, the 50 realizations of each model are compared to the reference rain type time series (Fig. 6 and Fig. 7). Focusing first on the ability of both models to simulate the occurrence of rain (i.e. without distinction between rain types), Fig. 6 compares the observed and simulated frequencies of rain occurrence for each season of the validation period 2001-2017. The results show that the non-parametric model properly simulates the overall proportion of rain (ratio simulated/observed rain frequency = 0.93), while the parametric model tends to underestimate rain occurrence (ratio simulated/observed rain frequency = 0.61). Focusing on the chronology of rain occurrence, it appears that the non-parametric model better reproduces the inter-annual variability of rain occurrence (correlation between observed and simulated time series=0.6) than the parametric model (correlation=0.4). Hence, according to the two above evaluation metrics, the non-parametric model outperforms the parametric one in terms of rain occurrence simulation. This can be explained by the fact that the relationships between the meteorological covariates and the presence of rain are probably more complex than the linear relationship assumed in the non-homogeneous Markov chain formulation of the parametric model (Eq.1). This hypothesis is reinforced by the fact that simulations driven by daily mean temperature only do not generate a dry bias (see Appendix A). However, although temperature is sufficient to capture the seasonality of rain type occurrence, using this covariate alone does not allow capturing the inter-annual variability of rain occurrence (see Appendix A) and therefore fails to preserve climate coherence when simulating rain types. Hence, it is preferred to keep the whole set of covariates to drive the parametric model, despite the resulting dry bias which constitutes the main drawback of the this model. In contrast, the non-parametric model can handle the complexity of the climate – rainfall relationships, mostly because the resampling approach is appropriate to reproduce complex and strongly non-linear relationships between the target variable and covariates (Mariethoz and Caers, 2015).

Focusing next on the simulation of rain types, Fig. 7 and Table 1 assess the ability of both models to reproduce the typical features of rain type occurrence conditional to the presence of rain highlighted in Sect. 2, namely seasonality, persistence and transition. Figure 7a shows that both models reproduce well the seasonality, with a correlation between observed and simulated





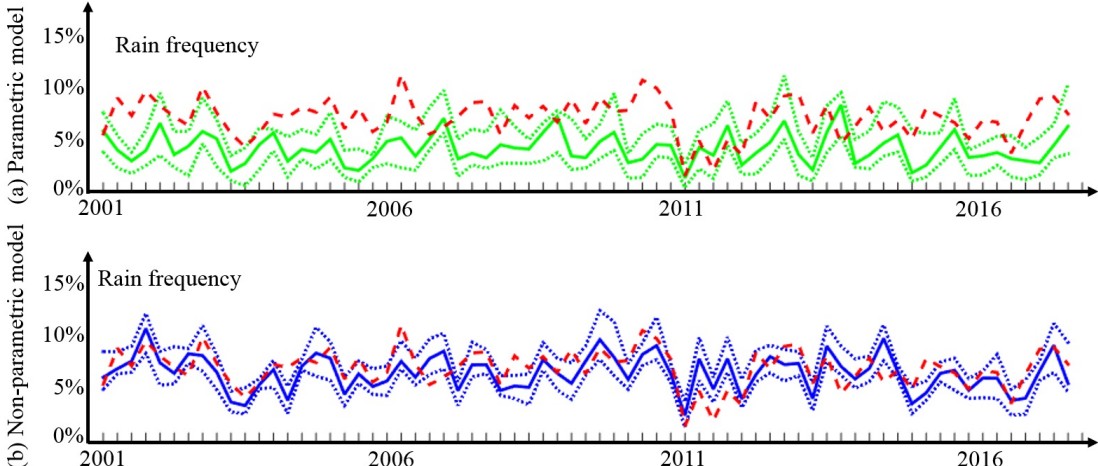

**Figure 6.** Rain occurrence frequency simulated by cross-validation and evaluated on a seasonal basis (seasons are DJF, MAM, JJA and SON). (a) Results for the parametric model (green). (b) Results for the non-parametric model (blue). In both graphs the dashed red line denotes the observed frequency. In the simulation results, continuous lines represent the median of the simulated ensembles (50 realizations), and dashed lines represent the Q10 and Q90 quantiles.

seasonality of 0.77 for the non-parametric model and 0.65 for the parametric model (average over all rain types except the very uncommon type 2 that is poorly simulated, see table 1). As for the simulation of rain occurrence, and for similar reasons, the non-parametric model outperforms the parametric one when simulating the occurrence of individual rain types. In contrast, Fig. 7b shows that the parametric model slightly outperforms the non-parametric model for simulating the persistence of rain

types. Indeed, the Kolmogorov–Smirnov statistic that measures the maximal divergence between the CDFs of rain type duration in observations and simulations is 0.12 for the parametric model and 0.13 for the non-parametric model (average over all rain types except type 2). The better performance of the parametric model regarding persistence was expected, since this feature is explicitly embedded in the model through the semi-Markov approach. Finally, Fig. 7c assesses the probability of transition between rain types, which is the main driver of rain type succession. Results show that for this metric, the non-parametric

model outperforms the parametric model with a Frobenius distance (i.e. L2 norm) between observed and simulated transition matrices of 0.03 in the non-parametric case and 0.10 in the parametric case (average over all rain types except type 2).

Overall, the non-parametric model outperforms the parametric model for the stochastic simulation of rain type occurrence. This better performance is linked to the ability of the non-parametric model to accurately reproduce the complex relationships that exist between the covariates and rain type occurrence. This property is essential to preserve the climatological coherence

for stochastic weather generation, and this is why the non-parametric model is preferred to the parametric one when performing RCM downscaling in the following case study. However, it should be noted here that the lower performance of the parametric





| Rain type | Correlation of monthly frequency of occurrence | | Kolmogorov-Smirnov statistic of duration CDF | | Frobinius distance of the matrix of transition | |
|---|---|---|---|---|---|---|
| | Parametric | MPS | Parametric | MPS | Parametric | MPS |
| 1 | **0.95** | 0.92 | 0.17 | **0.09** | 0.05 | **0.01** |
| 2 | -0.23 | **0.21** | 0.22 | **0.13** | 0.04 | **0.02** |
| 3 | 0.41 | **0.76** | 0.14 | **0.11** | 0.03 | **0.01** |
| 4 | 0.25 | **0.40** | **0.10** | 0.14 | 0.09 | **0.04** |
| 5 | 0.77 | **0.88** | **0.08** | 0.14 | 0.12 | **0.05** |
| 6 | **0.91** | 0.90 | **0.10** | 0.15 | 0.20 | **0.04** |

**Table 1.** Evaluation metrics for the 3 properties of rain type time series assessed in Fig. 7. For each case, the score of the best performing model is highlighted in bold. The first metric is the Pearson correlation between the observed and simulated frequency of rain type occurrence computed on a monthly basis, which evaluates the reproduction of rain type seasonality in simulations. The second metric is the Kolmogorov-Smirnov distance between the observed and simulated CDF of rain type duration, which quantifies the reproduction of rain type persistence in simulations. The third metric is the Frobinius distance between the observed and simulated matrix of transition, which evaluates the reproduction of rain type succession in simulations.

model is due to the complexity of the problem at hand, rather than a deficiency in the model itself, which extends to high temporal resolution problems some approaches that are state-of-the-art in daily resolution stochastic weather generators.

## 4.2 Application to RCM precipitation downscaling

For illustration purposes, the non-parametric model is used hereafter to simulate the evolution of rain type occurrence in a changing climate simulated by one RCM run extracted from the EURO-CORDEX climate downscaling experiment (Jacob et al., 2014). To drive the simulation of rain types in a changing climate, the same set of meteorological covariates as the one used for cross-validation is derived from one RCM run, more precisely from the Regional Atmospheric Climate MOdel of the Dutch national weather service (RACMO-KNMI (Van Meijgaard et al., 2008)) driven by the CNRM-CM5 Earth system model (Voldoire et al., 2013) forced according to the RCP8.5 emission scenario. Three intervals of 20 years each are selected to investigate the evolution of rain type occurrence over the $21^{st}$ century: 1997-2017 (reference period that encompasses the 2001-2017 calibration period for which rain type observations are available), 2037-2057, and 2077-2097. For each period, the meteorological covariates are extracted from the RCM simulation, averaged over the area of interest, and disaggregated at a 10 min resolution as described in Sect. 2. In addition, RCM outputs are bias-corrected using the CDF-t method for each variable separately (Vrac et al., 2012). After bias-correction of RCM data, the performance of the non-parametric model to simulate rain types in the present climate is almost identical for meteorological covariates derived from the RACMO-KNMI RCM and the ones derived from the ERA-5 reanalysis (see Appendix B).

For each 20-year period, 50 realizations are simulated using the non-parametric resampling approach presented in Sect. 3.2. To evaluate the projected changes in rain type distribution, Fig. 8 displays the evolution of the monthly frequency of rain type occurrence between the reference period 1997-2017 and the two future periods 2037-2057 and 2077-2097. Observed





**Figure 7.** Results of the cross-validation experiment for the simulation of rain types conditional to the presence of rain. (a) Seasonality of rain type occurrence, (b) rain type persistence, and (c) probability of transition between rain types. Observations are in red, simulations from the parametric model in green, and simulations from the non-parametric model in blue. In (a) and (b) continuous lines represent the median of the simulated ensembles (50 realizations), and dashed lines represent the Q10 and Q90 quantiles.

changes in rain type occurrence frequency are considered as significant if they exceed the uncertainty of the projection that is defined as the Q10-Q90 interval of the 50 realizations. It should be noted that in contrast to rain type occurrence, the simulated persistence and transition behavior of rain types remain constant over the whole test period (see Appendix C), and are not further considered hereafter. Results in Fig. 8 show that the frequency of rain occurrence slightly decreases in summer and increases in winter, spring and autumn. Among these changes, only the increase of rain occurrence during autumn and winter is significant. The distribution of rain types is more significantly modified than the occurrence of rain. More precisely, during winter, rain type 1 (stratiform) significantly declines while the frequency of rain type 3 (frontal) and 5 (moderately convective) significantly increases. During spring and fall, rain types 1 (stratiform) significantly declines while rain types 5, 6 (convective)





tend to increase but not significantly. Finally, during summer, rain types 1, 4 (stratiform) and 5 (moderately convective) decrease while rain types 3 (frontal) and 6 (strongly convective) increase, most of these changes being significant. Overall, it appears from this exploratory study that under the assumption of the specific RCM run used to simulate the meteorological covariates, convective and frontal rains could become more frequent at the expense of stratiform rains by the end of the $21^{st}$ century. The

most significant changes are obtained during winter and summer. A similar pattern of changes is simulated by the parametric model (see Appendix D), despite the persistence of the dry bias identified in Sect. 4.1. The similarity in rain type distribution produced by two models of different nature tends to indicate that the changes in meteorological covariates simulated by the RCM lead to a robust change in rain type distribution. It is worth mentioning that the evolution of rain behavior along the $21^{st}$ century simulated in the present study is qualitatively in line with results obtained over Western Europe by studies using

physical models, which anticipate more frequent heavy rains driven by convection or active fronts (Molnar et al., 2013; Faranda et al., 2019) at the expense of low intensity stratiform precipitations.

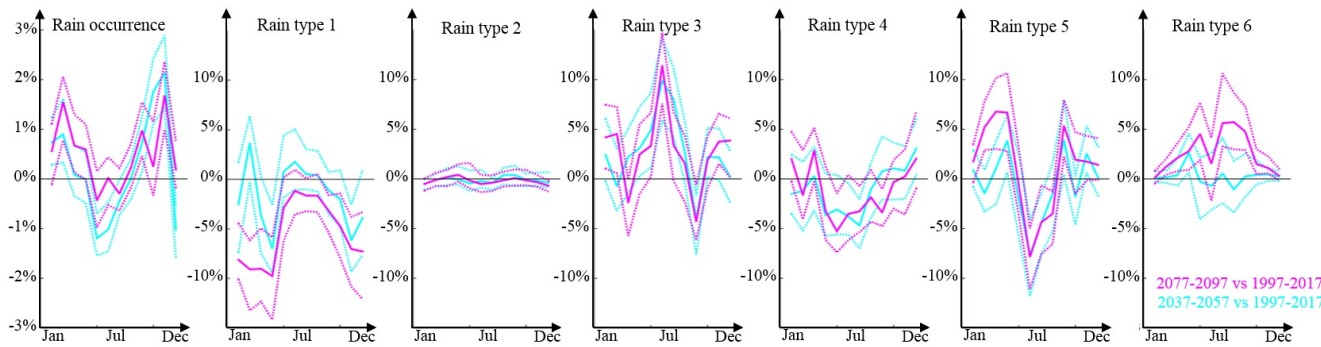

**Figure 8.** Changes in rain occurrence frequency (left panel) and rain type distribution (other panels) simulated using the non-parametric model: 2037-2057 vs 1997-2017 (light blue) and 2077-2097 vs 1997-2017 (purple). Continuous lines represent the median of the simulated ensembles (50 realizations), and dashed lines represent the Q10 and Q90 quantiles.

## 5   Concluding remarks

### 5.1   Discussion

By introducing a step of rain type simulation in the framework of stochastic weather generators, we suggest that for high

temporal resolution applications, the simulation of rain can be split in two steps. In a first instance, rain types are simulated conditional to meteorological covariates to account for the diversity of rain storms at the regional scale. This first step is the main focus of the present paper. For subsequent applications, we assume that the rain intensity can be simulated conditional to rain types. This is the classical aim of space-time distributed stochastic rainfall generators, which are becoming more and more common to address the needs of high-resolution hydrometeorological impact studies (Leblois and Creutin, 2013; Paschalis

et al., 2013; Nerini et al., 2017; Benoit et al., 2018a).





Hence, two main applications can be considered for the stochastic rain type simulation. The first one, briefly illustrated in Sect. 4.2, consists of assessing the evolution of the statistical signature of rainfall in a changing climate simulated by RCMs. It is worth noting that if one want to carefully evaluate the change in rain type occurrence that may emerge in the future, one should rely on a large ensemble of RCM-GCM-Emission scenarios combinations to properly capture the uncertainty on

meteorological covariates. In addition, one should keep in mind that the present approach only accounts for changes in the distribution of existing rain types, and therefore ignores the possible emergence of new rain types in response to climate conditions that have never been observed over the area of interest. Such new rain types could potentially by modelled by reparametrizing the stochastic rainfall model used to simulate local rain fields (Peleg et al., 2019) or by using rain analogs from areas that experience today the climate that is simulated in the future over the area of interest (Hallegatte et al., 2007; Fitzpatrick

and Dunn, 2019). The development of a framework to model emerging rain types is however left for future research.

The second application is the simulation of rain intensity at high space-time resolution while preserving the climatological coherence with covariates such as temperature, pressure, humidity and wind. As mentioned in the introduction, simulating rain intensity would require setting up and calibrating a high-resolution stochastic rainfall model for each rain type over the area of interest. Two advantages are expected to emerge from adding a rain type simulation step into stochastic rainfall modelling:

first, a relatively low number of rain types can be specified, which implies that the model of rain intensity has to be calibrated a limited number of times. This ensures that enough observations are available to calibrate the rain intensity model for each rain type, and therefore prevents model overfitting. The second advantage is the added flexibility to simulate rain storm dynamics, which allows to generate intra-storm variations of the space-time rainfall statistics.

## 5.2   Outlook

In this paper, two different methods have been proposed and thoroughly tested for the simulation of rain types time series conditional to meteorological covariates: a parametric approach based on a non-homogeneous semi-Markov model and a non-parametric approach based on the resampling of historical records using multiple-point statistics. Both methods have been assessed by cross-validation using a 17-year long rain type dataset in a mid-latitude climate (central Germany). Evaluation results favor the non-parametric approach because it better reproduces the complex rain type - climate relationships existing

in our test dataset. After validation, stochastic rain type simulation is applied to the downscaling of RCM projections over the $21^{st}$ century. Rain type simulations conditioned to meteorological covariates simulated by a Regional Climate Model under an RCP8.5 emission scenario indicate a possible change in rain type distribution by the end of the century, with an increased frequency of heavy rains driven by convection or active fronts, and a decline of low intensity stratiform precipitations.

The ability of stochastic simulations to generate realistic rain type time series when conditioned to meteorological covariates

advocates for including stochastic rain type simulation into weather generators in order to: (1) reproduce the internal variability or rain type occurrence, in particular inter-annual variability, seasonality, persistence and inter-type transitions, and (2) preserve the climatological coherence between rain statistics and meteorological covariates, in the present case temperature, pressure, humidity and wind. The above features make stochastic rain type simulation a convenient tool to account for the





non-stationnarity of rain statistics in the context of stochastic weather generators. This opens the door to sub-daily stochastic downscaling of climate projections, and rainfall simulations.

*Competing interests.* The authors declare that they have no conflict of interests.

*Acknowledgements.* All data and codes used in this study are open source and freely available. They can be found in the following reposito-
5  ries:

  – Radar data: https://opendata.dwd.de/climate_environment/CDC/grids_germany/5_minutes/radolan.

  – Rain type data: https://github.com/LionelBenoit/Stochastic_Raintype_Generator/Raintype_data.

  – Rain typing software: https://github.com/LionelBenoit/Rain_typing.

  – Stochastic rain type models: https://github.com/LionelBenoit/Stochastic_Raintype_Generator/codes.

10  – MPS simulation software: https://github.com/GAIA-UNIL/G2S.





## Appendix A:  Cross-validation of the parametric model using the daily mean temperature as only covariate

To test the hypothesis that 'complex relationships between rain type occurrence and meteorological covariates generate a dry bias when the parametric model is used', the cross-validation experiment presented in section 4.1 has been repeated using a simpler set of covariates. Here we illustrate this using daily mean temperature as the only covariate. Daily mean temperature

5  has been selected because of its strong seasonality that is expected to help reproducing the seasonality in rain type occurrence. Results show that using temperature only as covariate reduces the dry bias (Fig A1) and slightly improves the reproduction of seasonality and transition probabilities in the parametric setting (Fig A2). However, the inter-annual variability is not captured (Fig A1) probably because temperature is not sufficient to describe the state of the atmosphere that triggers the occurrence of the different rain types.

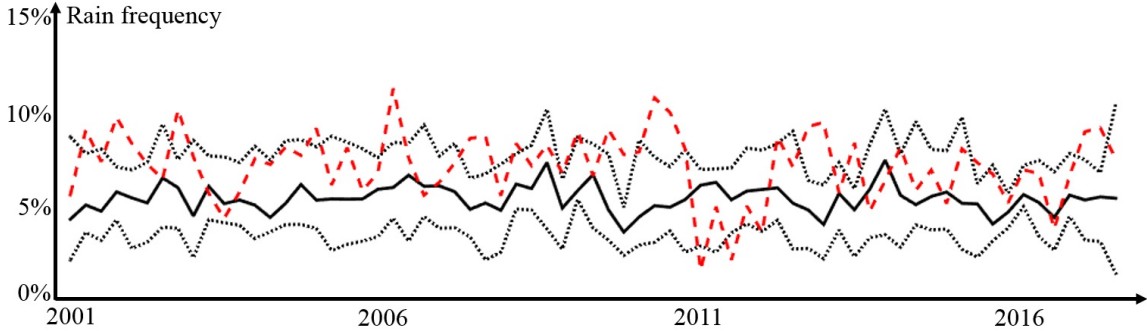

**Figure A1.** Rain occurrence frequency simulated by cross-validation and evaluated on a seasonal basis (seasons are DJF, MAM, JJA and SON), using the parametric model and daily mean temperature as unique covariate to drive rain type simulation. The dashed red line denotes the observed frequency, while the black curves correspond to simulated values. In simulation results, continuous lines represent the median of the simulated ensembles, and dashed lines represent the Q10 and Q90 quantiles.

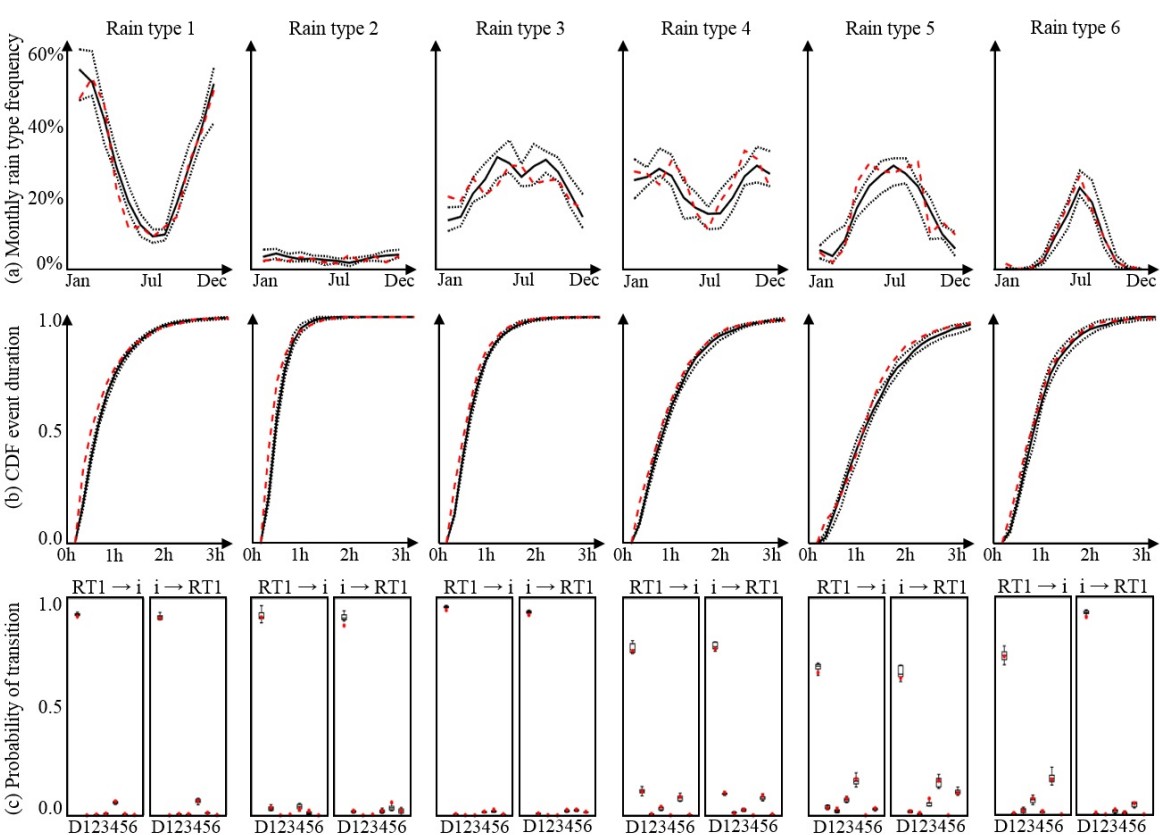

**Figure A2.** Results of the cross-validation experiment for the simulation of rain types conditional to the presence of rain, using the parametric model and daily mean temperature as only covariate. (a) Seasonality of rain type occurrence, (b) rain type persistence, and (c) probability of transition between rain types. Observations are in red and simulations are in black. In (a) and (b) continuous lines represent the median of the simulated ensembles, and dashed lines represent the Q10 and Q90 quantiles.





## Appendix B: RCM bias correction and performance of associated rain type simulations

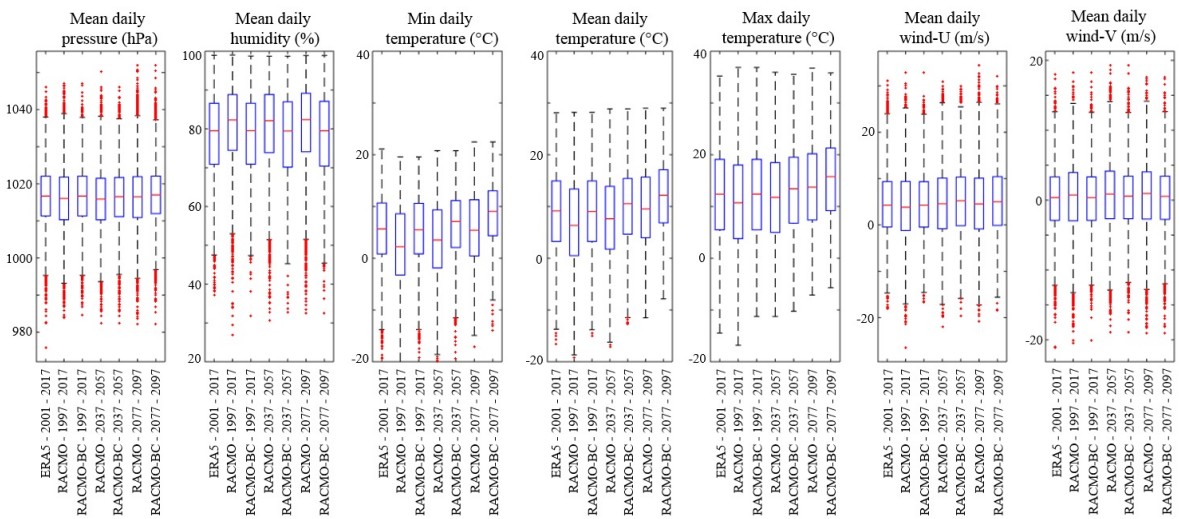

**Figure B1.** Evolution of the meteorological covariates along the different periods tested in this study: 2001-2017 (present climate, cross-validation), 1997-2017 (present climate, application to RCM downscaling), 2037-2057 and 2077-2097 (future climate, application to RCM downscaling). ERA5 refers to meteorological data derived from the ERA-5 reanalysis that are used for calibration and cross-validation. RACMO refers to meteorological data derived directly from the RACMO – CNRM regional climate model. RACMO-BC is similar to RACMO but has been bias-corrected using the CDF-t method and ERA5 data as reference for the present climate. RACMO-BC is the dataset that is used to condition the rain type simulations in subsection 4.2. In the boxplots the central red line denotes the median, the blue box encompasses the quantiles 25% to 75%, the whiskers encompass all non-outliers data points (i.e. +/- 2.7σ), and the red crosses denote outliers.

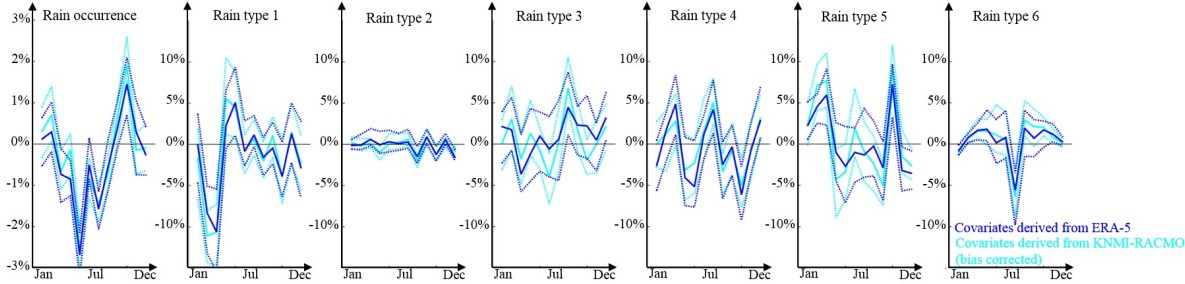

**Figure B2.** Performance of the non-parametric model to simulate rain type occurrence for the 2001-2017 period using two different sets of meteorological covariates: ERA-5 reanalysis (dark blue) and bias-corrected RACMO-KNMI RCM (light blue). This figure shows the difference between observed and simulated monthly rain occurrence (left panel) and rain type frequency (other panels). Continuous lines represent the median of the simulated ensembles (50 realizations), and dashed lines represent the Q10 and Q90 quantiles.



## Appendix C:  Evolution of rain type occurrence simulated by the non-parametric model

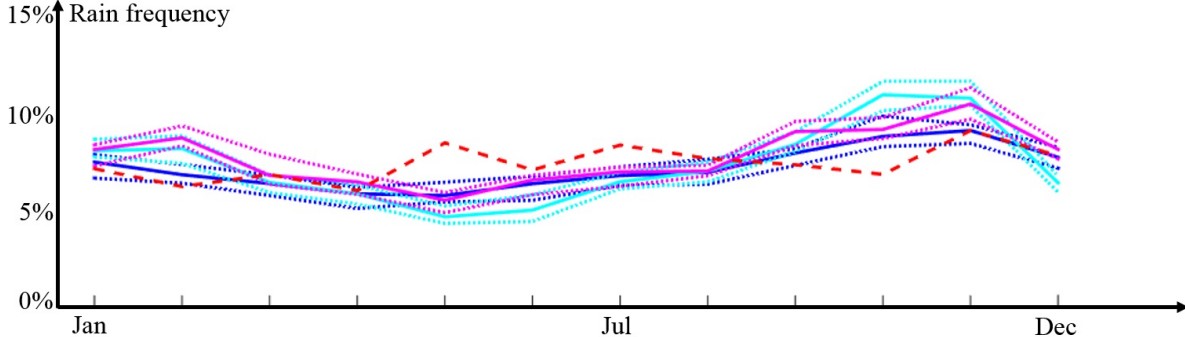

**Figure C1.** Change in rain occurrence frequency simulated using the non-parametric model. The dashed red line denotes observations (2001-2017), dark blue denotes simulation results for 1997-2017, light blue denotes simulation results for 2037-2057, and purple denotes simulation results for 2077-2097. In simulation results, continuous lines represent the median of the simulated ensembles, and dashed lines represent the Q10 and Q90 quantiles.



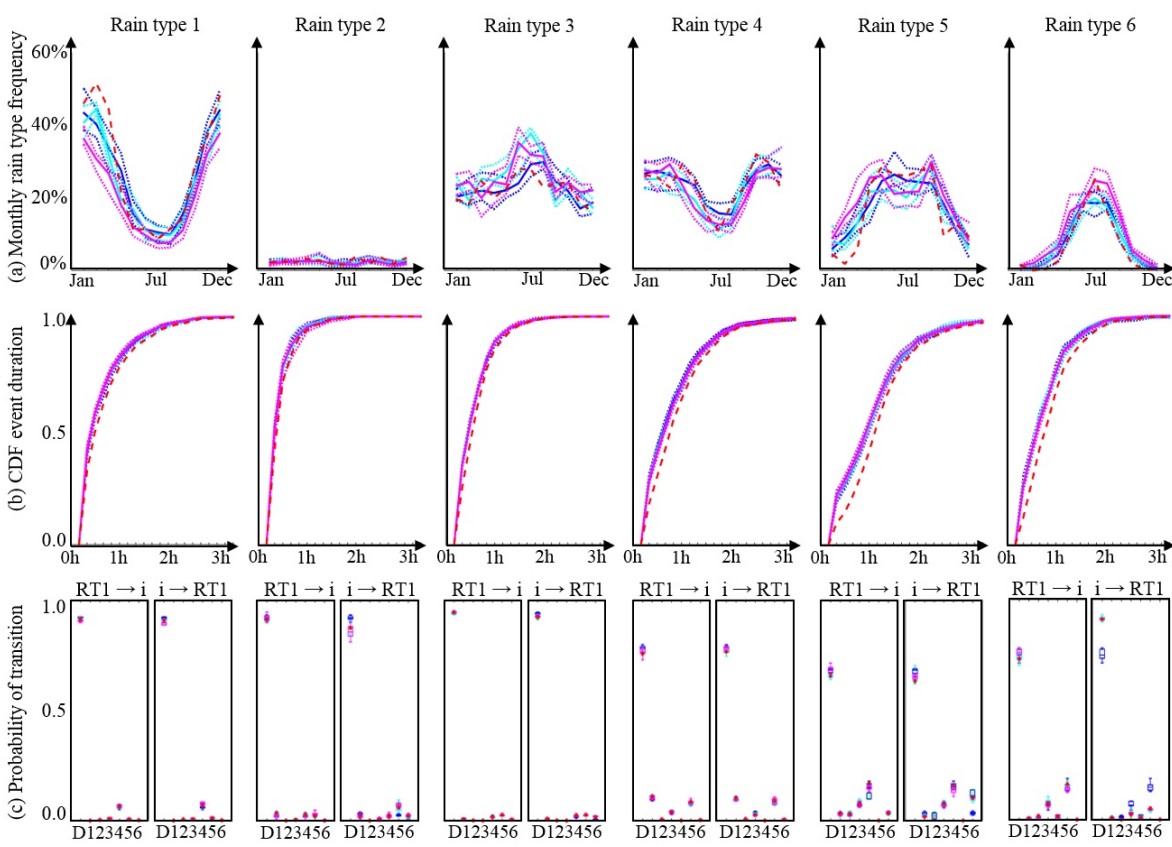

**Figure C2.** Changes in rain type distribution simulated using the non-parametric model. (a) Seasonality of rain type occurrence, (b) rain type persistence, and (c) probability of transition between rain types. Observations (2001-2017) are in red, simulations for 1997-2017 are in dark blue, simulations for 2037-2057 are in light blue, and simulations for 2077-2097 are in purple. In (a) and (b) continuous lines represent the median of the simulated ensembles, and dashed lines represent the Q10 and Q90 quantiles.





## Appendix D: Evolution of rain type occurrence simulated by the parametric model

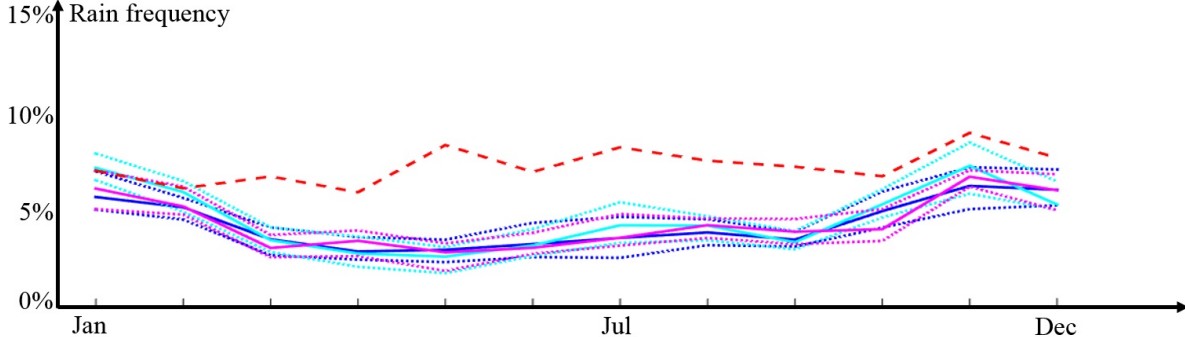

**Figure D1.** Change in rain occurrence frequency simulated using the parametric model. The dashed red line denotes observations (2001-2017), dark blue denotes simulation results for 1997-2017, light blue denotes simulation results for 2037-2057, and purple denotes simulation results for 2077-2097. In simulation results, continuous lines represent the median of the simulated ensembles, and dashed lines represent the Q10 and Q90 quantiles.

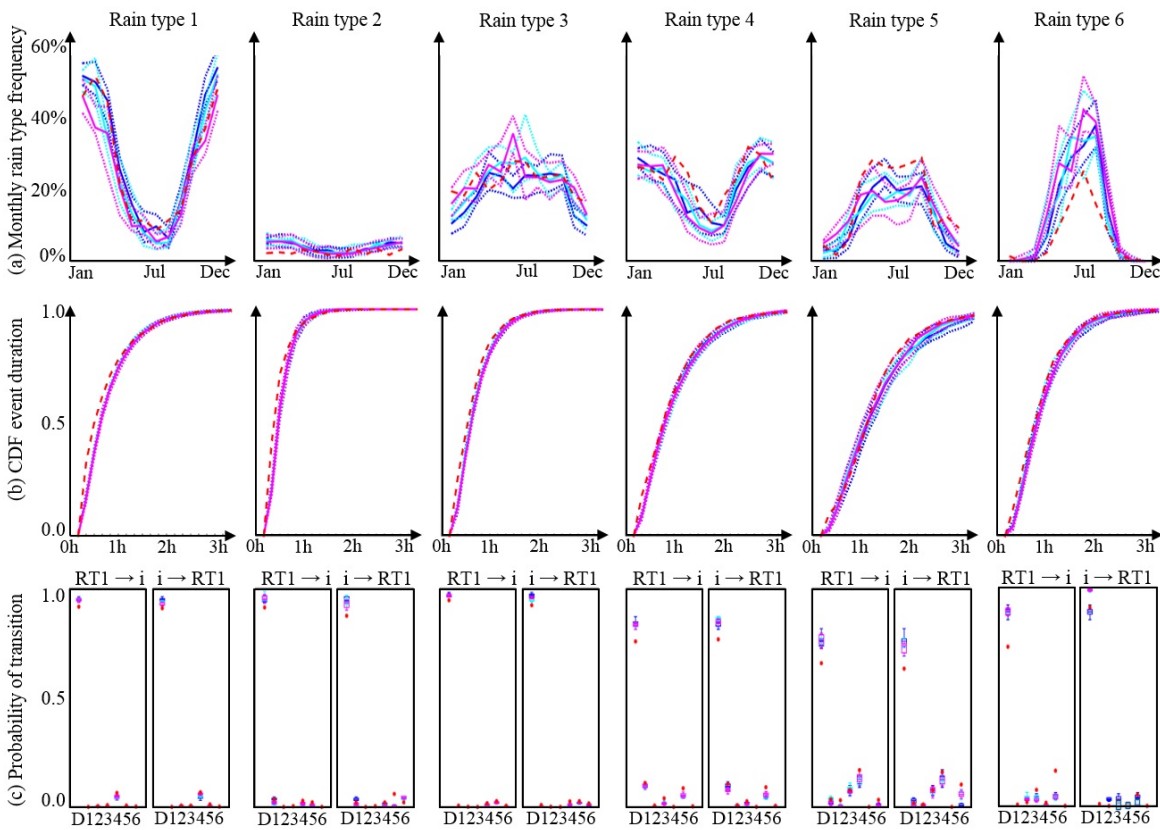

**Figure D2.** Changes in rain type distribution simulated using the parametric model. (a) Seasonality of rain type occurrence, (b) rain type persistence, and (c) probability of transition between rain types. Observations (2001-2017) are in red, simulations for 1997-2017 are in dark blue, simulations for 2037-2057 are in light blue, and simulations for 2077-2097 are in purple. In (a) and (b) continuous lines represent the median of the simulated ensembles, and dashed lines represent the Q10 and Q90 quantiles.

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
