# Peer review of "Accounting for rain type non-stationarity in sub-daily stochastic weather generators"

_Hydrology and Earth System Sciences, 2019_

## Referee Comment (RC1) · Anonymous Referee #1 · 30 Nov 2019

Review of Accounting for rain type non-stationarity in sub-daily stochastic weather generators by Benoit et al.

The paper presents a rain-type simulator conditional to meteorological covariate applied over central Germany. Two methods are compared and the authors show that the non-parametric approach outperforms the parametric one. In addition, the simulator is used to project changes in rain-type frequency and seasonality in future by application to RCM under RCP8.5.

I find the paper interesting, well-written and well-structured. My comments mainly touch some methodological issues and ask for some clarifications.

1) Rainy images are considered those with > 10% rainy pixels (what rain intensity threshold was applied to set a pixel as rainy?). I wonder if the selection of 10% as a threshold is important and may affect the conclusions? For example, if there is a distinct rain type with a small coverage area (<10%) we totally miss it. I understand a threshold needs to be set, but it should be clarified that there is no high sensitivity to this definition.

2) Radar data are not gauge-adjusted – this may affect the indices, especially the intensity indices. For example, the "Mean rain intensity over all rainy pixels" index is obviously affected by biases in the radar data that could be corrected with gauge adjustment. It can be claimed that such biases will be consistent and therefore will not affect classification, but biases in radar data could change along the years and thus the inter-annual frequency of rain types may be affected. I suggest addressing this point in the paper.

3) Rain type duration: please explain how is it computed? is it simply derived from consecutive maps series with the same type and does a single map with a different type end this series?

4) Long and short dry duration: is the 24h long/dry duration necessarily defined over days (i.e, midnight to midnight) or is it 24h with an arbitrary starting point? if the latter, how dry periods are split between "long" and "short" intervals?

5) It would be helpful to indicate the percent of rain amount out of total amount for each rain type, as well as for "dry" time-steps (i.e., with <10% rainy pixels).

6) Figure 2A: please indicate months so seasonality can be better realized

7) It would be helpful to mention statistical significance for the comparisons presented in Figure 3.

8) Eq. 1: I think that it should be written as: $P(S_t=j|S_{t-1}=i,x)=\ldots$ if not, please explain what is the source of index j on the right side of the equal sign.

9) Page 9, Lines 9-10: can you explain why different data are used for determining the Sigma and Mue matrices?

10) Fig. 6: can you explain the systematic negative bias in rain occurrence frequency for the two models?

---

## Referee Comment (RC2) · András Bárdossy (Referee) · 10 Jan 2020

*Review of the paper*
**Accounting for rain type non-stationarity in sub-daily stochastic weather generators**
*by Lionel Benoit, Mathieu Vrac, and Gregoire Mariethoz*
*submitted for publication in*
*Hydrology and Earth System Sciences*

The method of clustering rain types was presented in a previous paper. It is based on radar images, and thus not on real rainfall. Radar images are notoriously weak in quantifying precipitation amounts, which can lead to mismatches. Are the on ground rainfall characteristics really related to these types? The 10 % wet pixels for the rain classifications means that the beginning and the end of the events are neglected. This leads to a reduction of the durations. Wouldnt it be reasonable to apply a space time classification?

The choice of the meteorological covariates is not convincing. The seven variables seem to have weak relationships with the rain types. Are these variables really better if they are combined? A scatterplot of the variables with the indication of the rain types would be necessary to see if the variables are likely to explain the occurrence of the different rain types. Would not a similar typing of the spatial patterns of these variables be a better alternative to find a relationship? Another specific problem here is the use of daily covariates. Rainfall is often related to short time changes in temperature and air pressure. The suggested disaggregation procedure cannot cope with this and practically relates 10 min precipitation types to daily covariates through a pre-defined daily cycle. This of course reduces the possible influence of the covariates. Present observations could be used to see whether the covariates are better if available on higher resolution. The variable with the clearest signal is temperature, which may be the only reason why changes are detected in the RCM scenarios.

Simple year by year cross validation is not enough to show the applicability of the model for climate model downscaling. Instead a split sampling into dry and wet years and warm and cold years could help to know if the model is likely to handle climatic signals reasonably.

In my opinion the systematic bias of the parametric model indicates that it was not set up properly. Therefore, it would be important to modify it and remove the bias. I would certainly not try to apply a model which is biased.

Minor remarks:

1. Figure 2 upper panel: One cannot guess the fluctuations of the frequencies of the individual rain types except for type 1. I suggest to show individual lines instead.

2. Figure 7 (c): due to the very high dry transition probabilities the other transitions cannot be judged from this presentation.

In summary I think this paper contains a lot of interesting ideas, which deserve publication. On the other hand there are a lot problems which should be discussed before accepting it for publication. Therefore, I suggest a major revision.

---

## Referee Comment (RC3) · Anonymous Referee #3 · 25 Jan 2020

The manuscript suggests, through examples, that a division of rainfall time series into "rain types", based on radar images, can lead to a better performance of weather generators.

The claim promoted is that each suggested rain type is easily modelled using standard weather generators and that the sequence of rain types can be modelled without modelling the actual weather.

The authors need to take the study to the end and show that the suggested rain types are indeed modelled easier separately than together, and that this approach is actually operationally feasible. From Figures 2 and 5 it shows that rain types can transition directly to one another, meaning that having individual models for each rain type could be very challenging.

[Figure]

Also, the two stochastic rain types models compared both seem extremely cumbersome to work with, without showing really convincing results. Convincing results could only be to actually model not only the sequence of rain types, but go the step further and model the weather.

It is very unclear how the final results could actually look. Could this approach be used to make stochastic rainfall output resembling the radar data used for the rain type data? That would indeed be impressive. It would also be very interesting to compare such result to state-of-the-art 2D weather generators (like e.g. the AWE-GEN-2D).

As a standalone item, this manuscript is not very interesting. The fact that radar images can be used to make rain types has already been published elsewhere, and showing that you can get a 7-state Markov model to work is not very novel. You claim that it will improve stochastic modelling of rainfall, you need to also show it.

---

## Author Comment (AC1) · 13 Feb 2020

Dear Editor and Reviewers,

Thank you for your detailed comments and suggestions about our manuscript entitled "Accounting for rain type non-stationarity in sub-daily stochastic weather generators".

To capitalize on your propositions of improvement, we suggest to:

(1) Better explain how the proposed rain type simulation approach fits within the framework of stochastic rainfall simulation.

(2) Give more details about the selection of meteorological covariates.

(3) Better justify the 10% threshold selected to define a rainy radar image.

[Figure]

(4) Keep only the non-parametric model in order to ensure unbiased results, and to simplify the method section.

(5) Reinforce the validation section by adding a split-sampling test that demonstrates the ability of our model to handle climatic signals.

To this end, we consider to modify the plan of the paper as detailed hereafter. The content of the new sections is introduced together with our point-by-point responses to the comments of the reviewers, which are available in attached files. When several reviewers raised the same issue, we repeat our reply in each point-by-point response.

Hoping that our responses address your concerns, and that our propositions of improvements will fulfil your expectations,

Best regards,

Lionel Benoit, Mathieu Vrac and Gregoire Mariethoz.

———————————————————————————————

General overview of the proposed changes:

Considering the comments and recommendations of the reviewers, the outline of the paper will be amended as follow:

Non-stationary stochastic rain type generation: accounting for climate drivers [new title]

1. Introduction [new figure to contextualize our approach]

2. Example dataset of rain type time series and related meteorological covariates

2.1. Rain type time series [new material to justify the 10% threshold]

2.2. Meteorological covariates [new material to justify the joint use of covariates]

3. Stochastic rain type model [keep only the non-parametric model]

4. Model assessment

4.1. Cross-validation

4.2. Sensitivity to climate variability [new material]

5. Application to RCM downscaling

6. Concluding remarks

6.1. Discussion

6.2. Outlook

Appendix A: Cross-validation when accounting for low rain coverage [new material to support Sect 2.1]

Appendix B: Selection of meteorological covariates [new material to support Sect 2.2]

Appendix C: Markov-chain-based model of rain type occurrence [moved from Sect 3 to appendix]

Appendix D: RCM bias correction and performance of associated rain type simulations

Please also note the supplement to this comment:
https://www.hydrol-earth-syst-sci-discuss.net/hess-2019-562/hess-2019-562-AC1-supplement.pdf

**Supplement:**

Dear Reviewer#1,

Thank you for your detailed comments about our manuscript. All your suggestions have been considered, and we propose the following changes to address the questions you raised in your review.

In the following point-by-point responses, RC denotes a referee comment (in black), AR denotes the author response (in blue), and PM denotes the proposed modifications (in green).

Hoping that the proposed improvements will fulfill your expectations,

Best regards,

Lionel Benoit, Mathieu Vrac and Gregoire Mariethoz.
* * *
RC: Rainy images are considered those with > 10% rainy pixels (what rain intensity threshold was applied to set a pixel as rainy?). I wonder if the selection of 10% as a threshold is important and may affect the conclusions? For example, if there is a distinct rain type with a small coverage area (<10%) we totally miss it. I understand a threshold needs to be set, but it should be clarified that there is no high sensitivity to this definition.

AR: The threshold of 10% is indeed pretty high, and leads to classify around one third of rainy images in the dry type. To deal with this 'dry bias', the typing method we use to classify radar images actually incorporates a second step (Benoit et al, 2018). It aims at assigning to images with less than 10% rain coverage the type of the closest (along the time axis) classified image. Therefore we do not define a distinct rain type for images with a low rain coverage (<10%), but rather consider that this 'pseudo-type' is due to a transient behavior of rainfall at the onset and at the end of rain storms.

This point was overlooked in the early version of our manuscript, and we will therefore address it in details in the revised manuscript. In particular, a paragraph will be added in the main text to discuss how we deal with images with rain coverage <10%. And an appendix will be added to show that the performance of the proposed rain type simulation method is not degraded when images with less than 10% rain coverage are considered as dry.

PM:

New paragraph in section 2.1:

"*Using only radar images with more than 10% wet pixels to define rain types ensures a reliable classification, but at the cost of a dry bias (in the present dataset, 32% of the images measuring some rain have a rain fraction under 10%, and are therefore assigned to the dry type). To deal with images with less than 10% wet pixels, Benoit et al. (2018b) proposed to classify images with a small rain fraction (i.e. 0% < rain fraction < 10<%) in a second step by assigning them the type of the closest classified image (i.e. nearest neighbor interpolation in time). This post-processing scheme is not directly transferable to the context of simulation because no information about the previously misclassified images is available in simulation outputs. Two options can be considered to alleviate this problem. First, the rain type model defined in Sect 3 can be calibrated on the final classification (i.e. including images with low rain coverage), which results in simulations that preserve the actual rain proportion. However, using a classification that includes the beginning and the end of rain storms leads to less clear relationships between climate covariates and rain type occurrence, which may degrade simulation results. Hence the second option, which we follow in this paper that consists in (1) calibrating and running the rain type model for rain types defined only from radar images with more than 10% rain coverage, and next (2) re-adjusting the wet/dry balance by post-processing. The dry bias is corrected*

*assuming the ratio R = Ns/Nl between the number Ns of images with small rain coverage and the number Nl of images with large rain coverage as constant in observations and simulations. Subsequently, the number of epochs for which rain is simulated is increased by propagating the closest rain type to the RxNl dry epochs located at the beginning and at the end of rain storms. Appendix A shows that such post-processing performs well to mitigate the dry bias originating from the use of a 10% rain coverage threshold to define a wet image. However, since the present study focuses on climate - rain type relationships, which are better defined when considering only the first step of the classification, the aforementioned post-processing is not applied in the remainder of this paper. Hence, one should keep in mind that in the following the dry type also includes epochs with a small rain coverage (under 10%), and that a post-processing is required if the end-use is an application that involves the stochastic simulation of actual rain fields.*"

New appendix:

"*Appendix A: Cross-validation when accounting for small rain coverage.*

[Figure]

*Figure A1. Results of the cross-validation experiment when images with low rain coverage (rain fraction between 5% and 10%) are regarded as wet. (a) Seasonality of rain (and dry) type occurrence (Seasons are DJF (light blue), MAM (pink), JJA (red) and SON (yellow)), (b) rain type persistence, and (c) probability of transition between rain types. Observations are in red and are obtained by assigning the type of the closest classified image to epochs with rain fraction between 5% and 10%. Simulations are in blue and are obtained by propagating the closest rain type to the beginning and to the end of each rain event. In simulations, continuous lines represent the median of the simulated ensembles (50 realizations), and dashed lines represent the Q10 and Q90 quantiles.*"

RC: Radar data are not gauge-adjusted – this may affect the indices, especially the intensity indices. For example, the "Mean rain intensity over all rainy pixels" index is obviously affected by biases in the radar data that could be corrected with gauge adjustment. It can be claimed that such biases will be consistent and therefore will not affect classification, but biases in radar data could change along the years and thus the inter-annual frequency of rain types may be affected. I suggest addressing this point in the paper.

AR: This is true that changing radar biases could negatively affect the results of this study. Fortunately, the use of uniformly processed radar images minimizes the risk of drift in radar biases. In practice, we do not find any long-term trend in observed rain type frequencies. In addition, the results of the leave-one-year-out cross-validation procedure do not show any disagreement between climate-driven simulated rain types and observed rain types, which reflects the absence of outliers in observed rain types.
The above will be addressed in the first paragraph of Section 2.1 when we present the radar dataset.

PM: "*Using raw radar images can lead to biases in estimated rain intensities, but the impact of such biases on the classification are deemed negligible since the adopted approach focuses on rainfall space-time behavior rather than rainfall intensity. A more troublesome source of errors would be the change of radar biases along time, which could alter the inter-annual frequency of rain types. To alleviate this problem, uniformly reprocessed radar images are used as basis for the classification, which ensures a consistent data-cube throughout the period of interest. In practice, no adverse trend is noted in the observed rain type distribution (Fig. 3).*"

RC: Rain type duration: please explain how is it computed? is it simply derived from consecutive maps series with the same type and does a single map with a different type end this series?

AR: This is correct. The following description of rain type duration will be added in the revised manuscript (Sect. 2.1).

PM: "*Here rain type duration is defined as the duration (i.e. length along the time axis) of a segment of rain type time series with constant type. Each curve in Fig.3a therefore corresponds to the probability that a rain event of a given type does not exceed the duration given in abscissa.*"

RC: Long and short dry duration: is the 24h long/dry duration necessarily defined over days (i.e, midnight to midnight) or is it 24h with an arbitrary starting point? if the latter, how dry periods are split between "long" and "short" intervals?

AR: It is 24h with an arbitrary starting point. A dry period is split in as many 'long' intervals as possible, and the remaining dry time is distributed into two 'short dry' intervals at the beginning and at the end of the 'long dry' period. For example if there in case of a 49h dry period, it is split into: 30min short dry + 2 x 24h long dry + 30min short dry.

The following clarification will be added in the description of the parametric model (please note that following comments of reviewer#2, the description of the parametric model will be moved in appendix).

PM: "*In practice, if a dry period exceeds 24h, it is split into as many 'long dry' spells as possible, and the remaining time is distributed into two 'short dry' spells of equal length at the beginning and at the end of the overall dry period.*"

RC: It would be helpful to indicate the percent of rain amount out of total amount for each rain type, as well as for "dry" time-steps (i.e., with <10% rainy pixels).

AR: This will be indicated in the figure that investigates the main features of rain type time series.

PM:

[Figure]

"*Figure 3. Main features of a rain type time series (2000-2017) observed over central Germany. (a) Frequency of rain type occurrence computed at a seasonal basis (Seasons are DJF (light blue), MAM (pink), JJA (red) and SON (yellow)). (b) CDF of event duration stratified by rain type. (c) Empirical matrix of transition probability between rain types.*"

RC: Figure 2A: please indicate months so seasonality can be better realized.

AR: Ok, this will be implemented.

PM: Seasonality will be indicated with a color code in all figures where relevant, see Fig 3a above as an example.

RC: It would be helpful to mention statistical significance for the comparisons presented in Figure 3.

AR: Following the comments of Reviewer#2 the parametric model will be moved to appendix and the comparison between the two models will be removed. Fig 3 will therefore disappear.

RC: Eq. 1: I think that it should be written as: $P(S_t=j|S_{t-1}=i,x)=\ldots$ if not, please explain what is the source of index $j$ on the right side of the equal sign.

AR: This is indeed a better formulation.

PM: The equation will be corrected accordingly.

RC: Page 9, Lines 9-10: can you explain why different data are used for determining the Sigma and Mue matrices?

AR: Our initial description was misleading, in fact the same data are used for determining the Sigma and Mue matrices. We will reformulate the description of these matrices as follow.

PM: "*More precisely, $\Sigma$ is the empirical covariance matrix and $\mu_{ij}$ is the empirical mean of the covariates for the time steps where the transition from $i$ to $j$ occurs.*"

RC: Fig. 6: can you explain the systematic negative bias in rain occurrence frequency for the two models?

AR: In our opinion the systematic negative bias is significant only for the parametric model. For the non-parametric model, the occurrence of dry and rain types is properly simulated. Regarding the dry bias in the parametric model, we believe that it is due to the difficulty of conditioning to covariates in the framework of semi-Markov models. The discussion about the performance of the parametric model will be modified as follows (in Appendix C).

PM: "*Figure C2 shows the results of the same cross-validation experiment than in Sect. 4.1, but applied to the parametric model described above. Results show that this model generates a strong dry bias (ratio simulated/observed rain frequency = 0.61) and does not properly capture the inter-annual variability of rain occurrence (correlation between observed and simulated time series = 0.4). This can be explained by the fact that the relationships between the meteorological covariates and the presence of rain are probably more complex than the linear relationship assumed in the non-homogeneous Markov chain formulation of the parametric model, and by the fact that semi-Markov models do not allow for an easy conditioning to continuous-time covariates. These hypotheses are reinforced by the fact that simulations driven by daily mean temperature only do not generate a dry bias (not shown here).*"

References: Benoit, L., Vrac, M. and Mariethoz, G.: Dealing with non-stationarity in sub-daily stochastic rainfall models, Hydrology and Earth System Sciences, 22, 5919-5933, doi:10.5194/hess-22-5919-2018, 2018.

---

## Author Comment (AC2) · 13 Feb 2020

Dear Editor and Reviewers,

Thank you for your detailed comments and suggestions about our manuscript entitled "Accounting for rain type non-stationarity in sub-daily stochastic weather generators".

To capitalize on your propositions of improvement, we suggest to:

(1) Better explain how the proposed rain type simulation approach fits within the framework of stochastic rainfall simulation.

(2) Give more details about the selection of meteorological covariates.

(3) Better justify the 10% threshold selected to define a rainy radar image.

[Figure]

(4) Keep only the non-parametric model in order to ensure unbiased results, and to simplify the method section.

(5) Reinforce the validation section by adding a split-sampling test that demonstrates the ability of our model to handle climatic signals.

To this end, we consider to modify the plan of the paper as detailed hereafter. The content of the new sections is introduced together with our point-by-point responses to the comments of the reviewers, which are available in attached files. When several reviewers raised the same issue, we repeat our reply in each point-by-point response.

Hoping that our responses address your concerns, and that our propositions of improvements will fulfil your expectations,

Best regards,

Lionel Benoit, Mathieu Vrac and Gregoire Mariethoz.

——————————————————————————————

General overview of the proposed changes:

Considering the comments and recommendations of the reviewers, the outline of the paper will be amended as follow:

Non-stationary stochastic rain type generation: accounting for climate drivers [new title]

1. Introduction [new figure to contextualize our approach]

2. Example dataset of rain type time series and related meteorological covariates

2.1. Rain type time series [new material to justify the 10% threshold]

2.2. Meteorological covariates [new material to justify the joint use of covariates]

3. Stochastic rain type model [keep only the non-parametric model]

4. Model assessment

4.1. Cross-validation

4.2. Sensitivity to climate variability [new material]

5. Application to RCM downscaling

6. Concluding remarks

6.1. Discussion

6.2. Outlook

Appendix A: Cross-validation when accounting for low rain coverage [new material to support Sect 2.1]

Appendix B: Selection of meteorological covariates [new material to support Sect 2.2]

Appendix C: Markov-chain-based model of rain type occurrence [moved from Sect 3 to appendix]

Appendix D: RCM bias correction and performance of associated rain type simulations

Please also note the supplement to this comment:
https://www.hydrol-earth-syst-sci-discuss.net/hess-2019-562/hess-2019-562-AC2-supplement.pdf

**Supplement:**

Dear Prof. András Bárdossy (Reviewer#2),

Thank you for your detailed comments about our manuscript. All your suggestions have been considered, and we propose the following changes to address the questions you raised in your review.

In the following point-by-point responses, RC denotes a referee comment (in black), AR denotes the author response (in blue), and PM denotes the proposed modifications (in green).

Hoping that the proposed improvements will fulfill your expectations,

Best regards,

Lionel Benoit, Mathieu Vrac and Gregoire Mariethoz.

RC: Radar images are notoriously weak in quantifying precipitation amounts, which can lead to mismatches. Are the on ground rainfall characteristics really related to these types?

AR: This is correct that radar images are weak in quantifying precipitation amount; this is why we propose here to use them as a proxy of rain storm behavior, hence the concept of rain types. In case ground rainfall characteristics are the main focus of the application at hand, a stochastic rainfall model should therefore be calibrated (conditional to rain types) using more relevant rain estimates, e.g. in-situ observations or gauge-adjusted radar images. When combined with a properly set-up stochastic rainfall model, rain types have been shown to improve simulation results (Benoit et al, 2018). Ground rainfall characteristics are therefore related to rain types derived from radar images, but a second step of rain intensity simulation is needed to quantify the actual rain intensity at the ground level. To better explain the use of rain type simulation within the framework of stochastic rainfall simulation the last paragraph of the introduction will be improved as follows, with the addition of a new figure.

PM: "In this context, the main goal of this paper is to propose a new approach to leverage the use of rain types for encoding rain non-stationarity in the framework of stochastic weather generators. However, the finality differs from that of classical weather generators (Richardson, 1981; Wilks and Wilby, 1999; Peleg et al., 2017) since we aim at simulating rainfall conditional to already known meteorological covariates, instead of simulating jointly the whole weather (i.e., all variables). More precisely, we develop a method for stochastic simulation of rain type time series conditional to the current state of the atmosphere, i.e. conditional to meteorological variables such as pressure, temperature, humidity or wind (Fig 1a). These meteorological covariates are assumed to be known beforehand, either from observations, numerical weather model outputs, or from other stochastic simulations. The advantage of the proposed approach is twofold: firstly, using a stochastic simulation to generate rain types allows to properly reproduce the natural variability of rain type occurrence, and thereby to indirectly model the non-stationarity of rain statistics observed in historical datasets. Secondly, the conditioning of the stochastic rain type model to the state of the atmosphere preserves the relationships between rain type occurrence and climatological drivers. Once realistic rain type time series have been simulated (i.e. the core of this study, Fig 1a), high-resolution rain fields can be simulated conditional to rain types using any high-resolution stochastic rainfall generator (e.g., Vischel et al., 2011; Leblois and Creutin, 2013; Paschalis et al., 2013; Nerini et al., 2017; Benoit et al., 2018a) as illustrated in Fig 1b. Using rain types to guide the stochastic generation of synthetic rains has been shown to improve the realism of the resulting high-resolution space-time simulations (Benoit et al., 2018b).

---

## Author Comment (AC3) · 13 Feb 2020

Dear Editor and Reviewers,

Thank you for your detailed comments and suggestions about our manuscript entitled "Accounting for rain type non-stationarity in sub-daily stochastic weather generators".

To capitalize on your propositions of improvement, we suggest to:

(1) Better explain how the proposed rain type simulation approach fits within the framework of stochastic rainfall simulation.

(2) Give more details about the selection of meteorological covariates.

(3) Better justify the 10% threshold selected to define a rainy radar image.

(4) Keep only the non-parametric model in order to ensure unbiased results, and to simplify the method section.

(5) Reinforce the validation section by adding a split-sampling test that demonstrates the ability of our model to handle climatic signals.

To this end, we consider to modify the plan of the paper as detailed hereafter. The content of the new sections is introduced together with our point-by-point responses to the comments of the reviewers, which are available in attached files. When several reviewers raised the same issue, we repeat our reply in each point-by-point response.

Hoping that our responses address your concerns, and that our propositions of improvements will fulfil your expectations,

Best regards,

Lionel Benoit, Mathieu Vrac and Gregoire Mariethoz.

———————————————————————————————

General overview of the proposed changes:

Considering the comments and recommendations of the reviewers, the outline of the paper will be amended as follow:

Non-stationary stochastic rain type generation: accounting for climate drivers [new title]

1. Introduction [new figure to contextualize our approach]

2. Example dataset of rain type time series and related meteorological covariates

2.1. Rain type time series [new material to justify the 10% threshold]

2.2. Meteorological covariates [new material to justify the joint use of covariates]

3. Stochastic rain type model [keep only the non-parametric model]

4. Model assessment

4.1. Cross-validation

4.2. Sensitivity to climate variability [new material]

5. Application to RCM downscaling

6. Concluding remarks

6.1. Discussion

6.2. Outlook

Appendix A: Cross-validation when accounting for low rain coverage [new material to support Sect 2.1]

Appendix B: Selection of meteorological covariates [new material to support Sect 2.2]

Appendix C: Markov-chain-based model of rain type occurrence [moved from Sect 3 to appendix]

Appendix D: RCM bias correction and performance of associated rain type simulations

Please also note the supplement to this comment:
https://www.hydrol-earth-syst-sci-discuss.net/hess-2019-562/hess-2019-562-AC3-supplement.pdf

**Supplement:**

Dear Reviewer#3,

Thank you for your detailed comments about our manuscript. All your suggestions have been considered, and we propose the following changes to address the questions you raised in your review.

In the following point-by-point responses, RC denotes a referee comment (in black), AR denotes the author response (in blue), and PM denotes the proposed modifications (in green).

Hoping that the proposed improvements will fulfill your expectations,

Best regards,

Lionel Benoit, Mathieu Vrac and Gregoire Mariethoz.
* * *
RC: The authors need to take the study to the end and show that the suggested rain types are indeed modelled easier separately than together, and that this approach is actually operationally feasible. From Figures 2 and 5 it shows that rain types can transition directly to one another, meaning that having individual models for each rain type could be very challenging.

AR: The interest of using rain types to improve stochastic rainfall generation has been demonstrated in a previous paper (Benoit et al, 2018). Hence, the main aim of the present study is rather to propose a new framework to simulate rain types conditional to meteorological covariates in order to ensure climate coherence. However, the reviewer is right when pointing out that the aim of the paper must be better stated. To this end, we propose to improve the introduction by adding a figure showing how the proposed rain type simulation method can be embedded into the framework of stochastic rainfall simulation. In particular, this new figure illustrates that realistic transitions can be simulated when a rain type switch occurs within a single rain storm. The end of the introduction will be revised as follows to better define the target of the paper.

PM [Same modification than the reply to Reviewer#2. We repeat it here for easier readability]: PM: "*In this context, the main goal of this paper is to propose a new approach to leverage the use of rain types for encoding rain non-stationarity in the framework of stochastic weather generators. However, the finality differs from that of classical weather generators (Richardson, 1981; Wilks and Wilby, 1999; Peleg et al., 2017) since we aim at simulating rainfall conditional to already known meteorological covariates, instead of simulating jointly the whole weather (i.e., all variables). More precisely, we develop a method for stochastic simulation of rain type time series conditional to the current state of the atmosphere, i.e. conditional to meteorological variables such as pressure, temperature, humidity or wind (Fig 1a). These meteorological covariates are assumed to be known beforehand, either from observations, numerical weather model outputs, or from other stochastic simulations. The advantage of the proposed approach is twofold: firstly, using a stochastic simulation to generate rain types allows to properly reproduce the natural variability of rain type occurrence, and thereby to indirectly model the non-stationarity of rain statistics observed in historical datasets. Secondly, the conditioning of the stochastic rain type model to the state of the atmosphere preserves the relationships between rain type occurrence and climatological drivers. Once realistic rain type time series have been simulated (i.e. the core of this study, Fig 1a), high-resolution rain fields can be simulated conditional to rain types using any high-resolution stochastic rainfall generator (e.g., Vischel et al., 2011; Leblois and Creutin, 2013; Paschalis et al., 2013; Nerini et al., 2017; Benoit et al., 2018a) as illustrated in Fig 1b. Using rain types to guide the stochastic generation of synthetic rains has been shown to improve the realism of the resulting high-resolution space-time simulations (Benoit et al., 2018b).*

[Figure]

*Figure 1. Overview of stochastic rain type generation (core of this study), and its application to simulate high-resolution synthetic rain fields whose statistical properties depend on meteorological conditions. (a) Rain type simulation framework developed in this study. (b) Illustration of stochastic rainfall simulation conditioned to changing rain types. In the bottom line of (a), the observed rain types are in red, and the gray shaded background denotes the probability of rain type occurrence derived from stochastic rain type simulations conditioned to the meteorological covariates displayed in the 4 top lines. In (b), the upper row displays actual rain fields observed by radar imagery, and the two bottom rows display two stochastic simulations of synthetic rain fields for the same period (Benoit et al., 2018a)."*

RC: Also, the two stochastic rain types models compared both seem extremely cumbersome to work with, without showing really convincing results.

AR: The parametric model was indeed a bit complex, and following a comment from Reviewer#2 it will be moved in appendix. This will hopefully make the methodological part easier to follow. Regarding the non-parametric model (based on Multiple-Point Statistics), it is nothing but a resampling of an historical dataset that preserves patterns. It is therefore quite easy to work with this model, especially because the software used to perform simulations is freely available with an extensive documentation (https://gaia-unil.github.io/G2S/). In the present application of rain type simulation, a basic call (i.e. without complicated options) of the G2S software allows to get all the results shown in the paper.

PM: The parametric model will be moved in appendix and overlooked in the main text. This will hopefully make the 'Section 3: Stochastic rain type model' simpler.

RC: Convincing results could only be to actually model not only the sequence of rain types, but go the step further and model the weather.

AR: The main aim of this study is to simulate rain types conditional to meteorological covariates, and therefore to pave the way to stochastic rainfall simulations which are coherent with the co-occurring climate conditions. Hence, the target is not to design a stochastic weather generator, but to focus on 'meteorologically realistic' rainfall simulations. We believe that the frequent mentions to 'weather generator(s)' throughout the first version of this paper were misleading. We will therefore remove most of these mentions in the revised manuscript, and modify the title to better state our objectives.

PM:

- Reduce the number of mentions to stochastic weather generators.

- Improve the introduction to better define the target of the paper [cf reply to comment#1 above].

- Change the title to: Non-stationary stochastic rain type generation: accounting for climate drivers.

RC: It is very unclear how the final results could actually look. Could this approach be used to make stochastic rainfall output resembling the radar data used for the rain type data? That would indeed be impressive.

AR: The new figure 1 shows that it is indeed possible to simulate synthetic radar images resembling the radar data used for model calibration. In addition, it will be better specified in the introduction that using a stochastic rainfall model conditioned to rain types has been proved to improve the final simulations (Benoit et al, 2018).

RC: It would also be very interesting to compare such result to state-of-the-art 2D weather generators (like e.g. the AWE-GEN-2D).

AR: The way synthetic radar images have been simulated in the new Fig 1b is almost identical to the rain generation module of AWE-GEN-2D (i.e. truncated and transformed multivariate Gaussian field, cf Benoit et al, 2018a, Peleg et al, 2017 and Paschalis et al, 2013). In that respect, very similar results are expected. The other aspect that could be compared is the realism of the relationships between meteorological covariates and rainfall simulations. However, the philosophy of the two approaches is fundamentally different, which prevents a fair comparison. In AWE-GEN-2D, rainfall is simulated first, and the other meteorological parameters are simulated conditional to rainfall. Here we do the opposite, and condition rain types (and in turn rain intensity) to the state of the atmosphere. This is justified by the different aims of the two approaches, the focus of the present paper being to propose a stochastic parametrization of rainfall conditional to climatic conditions.

RC: As a standalone item, this manuscript is not very interesting. The fact that radar images can be used to make rain types has already been published elsewhere, and showing that you can get a 7-state Markov model to work is not very novel. You claim that it will improve stochastic modelling of rainfall, you need to also show it.

AR: We hope the above answers convinced you that stochastic rain type simulation is an interesting topic, and can improve stochastic rainfall modelling. We also hope that clarifications about the aims of the paper, and how our contribution fits within the context of stochastic rainfall (and not weather) generators respond your inquiries about practical applications of the proposed approach.
* * *
References:

Benoit, L., Vrac, M. and Mariethoz, G.: Dealing with non-stationarity in sub-daily stochastic rainfall models, Hydrology and Earth System Sciences, 22, 5919-5933, doi:10.5194/hess-22-5919-2018, 2018.

Benoit, L., Allard, D. and Mariethoz, G.: Stochastic Rainfall Modelling at Sub-Kilometer Scale, Water Resources Research, 54, 4108-4130, doi:10.1029/2018WR022817, 2018a.

Paschalis, A., Molnar, P., Fatichi, S., and Burlando, P.: A stochastic model for high-resolution space-time precipitation simulation. Water Resources Research, 49, doi:10.1002/2013WR014437, 2013.

Peleg, N., Fatichi, S., Paschalis, A., Molnar, P. and Burlando, P.: An advanced stochastic weather generator for simulating 2-D high-resolution climate variables, Journal of Advances in Modeling Earth Systems, 9, 1595-1627, doi:10.1002/2016MS000854, 2017.

---

## Author Response (AR1)

Dear Editor and Reviewers,

Thank you for your detailed comments and suggestions about our manuscript entitled "Non-stationary stochastic rain type generation: accounting for climate drivers".

The paper has been revised accordingly. Please find hereafter the details of the changes in the form of an item-by-item response (in blue) to your comments (in black). Please note that the page and line numbers indicated hereafter refer to the ones of the revised manuscript.

Hoping that our responses address your concerns, and that our propositions of improvements will fulfil your expectations,

Best regards,

Lionel Benoit, Mathieu Vrac and Gregoire Mariethoz.
* * *
Responses to the comments of Reviewer #1:

RC: Rainy images are considered those with > 10% rainy pixels (what rain intensity threshold was applied to set a pixel as rainy?). I wonder if the selection of 10% as a threshold is important and may affect the conclusions? For example, if there is a distinct rain type with a small coverage area (<10%) we totally miss it. I understand a threshold needs to be set, but it should be clarified that there is no high sensitivity to this definition.

AR: The threshold of 10% is indeed pretty high, and leads to classify around one third of rainy images in the dry type. To deal with this 'dry bias', the typing method we use to classify radar images actually incorporates a second step (Benoit et al, 2018). It aims at assigning to images with less than 10% rain coverage the type of the closest (along the time axis) classified image. Therefore we do not define a distinct rain type for images with a low rain coverage (<10%), but rather consider that this 'pseudo-type' is due to a transient behavior of rainfall at the onset and at the end of rain storms.

This point was overlooked in the early version of our manuscript, and we therefore address it in details in the revised manuscript. In particular, a paragraph has been added in Sect. 2.1 to discuss how we deal with images with rain coverage <10% (p5, L6 – p6, L7). In addition, a new supplementary material (supplementary material 1) has been added to show that the performance of the proposed rain type simulation method is not degraded when images with less than 10% rain coverage are considered as dry.

RC: Radar data are not gauge-adjusted – this may affect the indices, especially the intensity indices. For example, the "Mean rain intensity over all rainy pixels" index is obviously affected by biases in the radar data that could be corrected with gauge adjustment. It can be claimed that such biases will be consistent and therefore will not affect classification, but biases in radar data could change along the years and thus the inter-annual frequency of rain types may be affected. I suggest addressing this point in the paper.

AR: This is true that changing radar biases could negatively affect the results of this study. Fortunately, the use of uniformly processed radar images minimizes the risk of drift in radar biases. In practice, we do not find any long-term trend in observed rain type frequencies. In addition, the results of the leave-one-year-out cross-validation procedure do not show any disagreement between climate-driven simulated rain types and observed rain types, which reflects the absence of outliers in observed rain types.

The above is now addressed in the first paragraph of Section 2.1 when we present the radar dataset (p4, L16-22).

RC: Rain type duration: please explain how is it computed? is it simply derived from consecutive maps series with the same type and does a single map with a different type end this series?

AR: This is correct. A better description of rain type duration has been added in the revised manuscript (p7, L2-4).

RC: Long and short dry duration: is the 24h long/dry duration necessarily defined over days (i.e, midnight to midnight) or is it 24h with an arbitrary starting point? if the latter, how dry periods are split between "long" and "short" intervals?

AR: It is 24h with an arbitrary starting point. A dry period is split in as many 'long' intervals as possible, and the remaining dry time is distributed into two 'short dry' intervals at the beginning and at the end of the 'long dry' period. For example, in case of a 49h dry period, it is split into: 30min short dry + 2 x 24h long dry + 30min short dry.

This has been clarified in the description of the parametric model (Supplementary material 3, p1, L12-14).

RC: It would be helpful to indicate the percent of rain amount out of total amount for each rain type, as well as for "dry" time-steps (i.e., with <10% rainy pixels).

AR: This is now mentioned in Fig 3.

RC: Figure 2A: please indicate months so seasonality can be better realized.

AR: Seasonality is now indicated by colors in Fig 3 and 6.

RC: It would be helpful to mention statistical significance for the comparisons presented in Figure 3.

AR: Following the comments of Reviewer#2 the parametric model has been moved to appendix and the comparison between the two models has been removed.

RC: Eq. 1: I think that it should be written as: P(St=j|St-1=i,x)=…. if not, please explain what is the source of index j on the right side of the equal sign.

AR: This is indeed a better formulation. We corrected it in supplementary material 3.

RC: Page 9, Lines 9-10: can you explain why different data are used for determining the Sigma and Mue matrices?

AR: Our initial description was misleading, in fact the same data are used for determining the Sigma and Mue matrices. The description of these matrices has been reformulated in supplementary material 3, p1, L23-25.

RC: Fig. 6: can you explain the systematic negative bias in rain occurrence frequency for the two models?

AR: In our opinion the systematic negative bias is significant only for the parametric model. For the non-parametric model, the occurrence of dry and rain types is properly simulated. Regarding the dry bias in the parametric model, we believe that it is due to the difficulty of conditioning to covariates in the framework of semi-Markov models. The discussion about the performance of the parametric model has been modified accordingly (Supplementary material 3, p1, L41 – p2, L3).

Responses to the comments of Reviewer #2:

RC: Radar images are notoriously weak in quantifying precipitation amounts, which can lead to mismatches. Are the on ground rainfall characteristics really related to these types?

AR: This is correct that radar images are weak in quantifying precipitation amount; this is why we propose here to use them as a proxy of rain storm behavior, hence the concept of rain types. In case ground rainfall characteristics are the main focus of the application at hand, a stochastic rainfall model should therefore be calibrated (conditional to rain types) using more relevant rain estimates, e.g. in-situ observations or gauge-adjusted radar images. When combined with a properly set-up stochastic rainfall model, rain types have been shown to improve simulation results (Benoit et al, 2018). Ground rainfall characteristics are therefore related to rain types derived from radar images, but a second step of rain intensity simulation is needed to quantify the actual rain intensity at the ground level. To better explain the use of rain type simulation within the framework of stochastic rainfall simulation the last paragraph of the introduction has been improved (p2, L19-34), with the addition of the new figure 1.

RC: The 10 % wet pixels for the rain classifications means that the beginning and the end of the events are neglected. This leads to a reduction of the durations.

AR [Same answer than the reply to Reviewer#1. We repeat it here for easier readability]: The threshold of 10% is indeed pretty high, and leads to classify around one third of rainy images in the dry type. To deal with this 'dry bias', the typing method we use to classify radar images actually incorporates a second step (Benoit et al, 2018). It aims at assigning to images with less than 10% rain coverage the type of the closest (along the time axis) classified image. Therefore we do not define a distinct rain type for images with a low rain coverage (<10%), but rather consider that this 'pseudo-type' is due to a transient behavior of rainfall at the onset and at the end of rain storms.

This point was overlooked in the early version of our manuscript, and we therefore address it in details in the revised manuscript. In particular, a paragraph has been added in Sect. 2.1 to discuss how we deal with images with rain coverage <10% (p5, L6 – p6, L7). In addition, a new supplementary material (supplementary material 1) has been added to show that the performance of the proposed rain type simulation method is not degraded when images with less than 10% rain coverage are considered as dry.

RC: Wouldn't it be reasonable to apply a space-time classification?

AR: The present classification already resorts to space-time statistics to define rain types, and is therefore implicitly a space-time classification. We added a sentence in the description of the classification method to better explain this point (p4, L16-17). However, the whole area of interest is always considered as covered by statistically homogeneous rain fields (i.e. constant rain type in space), which is indeed a strong assumption. Nevertheless, also considering non-stationarity of rain types in space would make the classification very difficult. In particular, moving neighborhoods would have to be defined to assess the spatial statistics over sub-domains. We therefore restrict our present work to a temporal classification of space-time statistics, hence assuming that the study area is small enough to assume spatially stationary rain fields at each time step. This is obviously a simplifying assumption, and one should keep in mind that this can generate abrupt changes in simulated rain fields as shown in the new Fig 1. This restricts the use of our method to areas of limited extend to ensure as much spatial stationarity as possible, as specified in section 2.1 *("We focus hereafter on a 100 km x 100 km squared area centered on the city of Jena in the Land of Thüringen, Germany (Fig. 2a). This area has been chosen because its flat topography and its location far from coastlines or major topographic barriers ensure spatially homogeneous rain fields, allowing to focus on the temporal component of rainfall non-stationarity.", p4, L8-10).* In case of large areas, or areas where a strong non-stationarity of rainfall in space is suspected (e.g. mountains or coastlines), a truly space-time classification should be considered.

RC: The choice of the meteorological covariates is not convincing. The seven variables seem to have weak relationships with the rain types. Are these variables really better if they are combined? A scatterplot of the variables with the indication of the rain types would be necessary to see if the variables are likely to explain the occurrence of the different rain types. Would not a similar typing of the spatial patterns of these variables be a better alternative to find a relationship?

AR: A new dedicated sub-section has been created to address the concerns raised above and to better emphasize on the importance of the chosen meteorological covariates (Sect 2.1: Meteorological covariates). In this section, we follow the advice of Reviewer#2 and study the joint influence of meteorological covariates on rain type occurrence (p7, L29-33 and Supplementary material 2, Figure 3). From the figure SI2-Fig3, it appears that if combined, the proposed covariates explain pretty well rain type occurrence. This is further reinforced by the ability of the proposed approach to capture the inter-annual variability of rain type occurrence, as highlighted for instance in figure 6. Finally, the use of weather types instead of a set of pointwise covariates is discussed in supplementary material 2, p1, L6-11.

RC: Another specific problem here is the use of daily covariates. Rainfall is often related to short time changes in temperature and air pressure. The suggested disaggregation procedure cannot cope with this and practically relates 10 min precipitation types to daily covariates through a pre-defined daily cycle. This of course reduces the possible influence of the covariates. Present observations could be used to see whether the covariates are better if available on higher resolution. The variable with the clearest signal is temperature, which may be the only reason why changes are detected in the RCM scenarios.

AR: We agree that rainfall is often related to short time changes in temperature and air pressure, and that the disaggregation procedure applied to meteorological covariates is therefore important to ensure high quality rain type simulations. As proposed by Reviewer#2, we compared our disaggregated covariates with their counterparts observed in-situ at high resolution by a weather station located within the area of interest. This is now discussed p7, L22-25, and in details in Supplementary material 2. In addition, we compared the performance of our rain type model when forced by (1) disaggregated covariates and (2) in-situ observations of the covariates. The results of this experiment are detailed in supplementary material 2 and mentioned in the main text p7, L22-25. These results show that the proposed disaggregation method performs well to capture sub-daily variations of the covariates. In addition, rain type simulations driven by the two covariate datasets (disaggregated vs high resolution in-situ) give very similar results. Consequently, we choose to keep the disaggregation of daily covariates to drive rain type simulations because this approach fits better with the illustration study considered in this paper.

RC: Simple year by year cross validation is not enough to show the applicability of the model for climate model downscaling. Instead a split sampling into dry and wet years and warm and cold years could help to know if the model is likely to handle climatic signals reasonably.

AR: We agree. A split-sampling has been added to the validation section (4.2: Sensitivity to climate variability, p11, L21 – p13, L2. The results of this split-sampling test show that our model properly captures the impact of the climatic signal on rain types.

RC: In my opinion the systematic bias of the parametric model indicates that it was not set up properly. Therefore, it would be important to modify it and remove the bias. I would certainly not try to apply a model which is biased.

AR: We agree that we should not apply a biased model, but we believe that the poor performance of the parametric model is not due to a wrong setting but rather to the inherent difficulty to condition a semi-Markov model to continuous-time covariates. This is now clearly stated in the revised manuscript (p8, L10-13). To avoid using a biased model, the parametric model has been removed from the main text and moved in supplementary material 3.

RC: Figure 2 upper panel: One cannot guess the fluctuations of the frequencies of the individual rain types except for type 1. I suggest to show individual lines instead.

AR: The figure (now Fig 3) has been revised accordingly.

RC: Figure 7 (c): due to the very high dry transition probabilities the other transitions cannot be judged from this presentation.

AR: The figure has been revised accordingly (in the revised manuscript it is figure 6).
* * *
Responses to the comments of Reviewer #3:

RC: The authors need to take the study to the end and show that the suggested rain types are indeed modelled easier separately than together, and that this approach is actually operationally feasible. From Figures 2 and 5 it shows that rain types can transition directly to one another, meaning that having individual models for each rain type could be very challenging.

AR: The interest of using rain types to improve stochastic rainfall generation has been demonstrated in a previous paper (Benoit et al, 2018). Hence, the main aim of the present study is rather to propose a new framework to simulate rain types conditional to meteorological covariates in order to ensure climate coherence. However, the reviewer is right when pointing out that the aim of the paper must be better stated. To this end, the introduction has been improved by adding the new figure 1 that shows how the proposed rain type simulation method can be embedded into the framework of stochastic rainfall simulation. In particular, this new figure illustrates that realistic transitions can be simulated when a rain type switch occurs within a single rain storm. In addition, the end of the introduction has been revised to better define the target of the paper (p2, L19-34).

RC: Also, the two stochastic rain types models compared both seem extremely cumbersome to work with, without showing really convincing results.

AR: The parametric model was indeed a bit complex, and following a comment from Reviewer#2 it has been moved to supplementary material 3. This hopefully makes the methodological part easier to follow. Regarding the non-parametric model (based on Multiple-Point Statistics), it is nothing but a resampling of an historical dataset that preserves patterns. It is therefore quite easy to work with this model, especially because the software used to perform simulations is freely available with an extensive documentation (https://gaia-unil.github.io/G2S/). In the present application of rain type simulation, a basic call (i.e. without complicated options) of the G2S software allows to get all the results shown in the paper.

RC: Convincing results could only be to actually model not only the sequence of rain types, but go the step further and model the weather.

AR: The main aim of this study is to simulate rain types conditional to meteorological covariates, and therefore to pave the way to stochastic rainfall simulations which are coherent with the co-occurring climate conditions. Hence, the target is not to design a stochastic weather generator, but to focus on 'meteorologically realistic' rainfall simulations. We believe that the frequent mentions to 'weather generator(s)' throughout the first version of this paper were misleading. We therefore removed most of these mentions in the revised manuscript, and modified the title of the paper to better state our objectives.

RC: It is very unclear how the final results could actually look. Could this approach be used to make stochastic rainfall output resembling the radar data used for the rain type data? That would indeed be impressive.

AR: The new figure 1 shows that it is indeed possible to simulate synthetic radar images resembling the radar data used for model calibration. In addition, it is now better specified in the introduction that using a stochastic rainfall model conditioned to rain types has been proved to improve the final simulations (Benoit et al, 2018).

RC: It would also be very interesting to compare such result to state-of-the-art 2D weather generators (like e.g. the AWE-GEN-2D).

AR: The way synthetic radar images have been simulated in the new Fig 1b is almost identical to the rain generation module of AWE-GEN-2D (i.e. truncated and transformed multivariate Gaussian field, cf Benoit et al, 2018a, Peleg et al, 2017 and Paschalis et al, 2013). In that respect, very similar results are expected. The other aspect that could be compared is the realism of the relationships between meteorological covariates and rainfall simulations. However, the philosophy of the two approaches is fundamentally different, which prevents a fair comparison. In AWE-GEN-2D, rainfall is simulated first, and the other meteorological parameters are simulated conditional to rainfall. Here we do the opposite, and condition rain types (and in turn rain intensity) to the state of the atmosphere. This is justified by the different aims of the two approaches, the focus of the present paper being to propose a stochastic parametrization of rainfall conditional to climatic conditions.

RC: As a standalone item, this manuscript is not very interesting. The fact that radar images can be used to make rain types has already been published elsewhere, and showing that you can get a 7-state Markov model to work is not very novel. You claim that it will improve stochastic modelling of rainfall, you need to also show it.

AR: We hope the above answers convinced you that stochastic rain type simulation is an interesting topic, and can improve stochastic rainfall modelling. We also hope that clarifications about the aims of the paper, and how our contribution fits within the context of stochastic rainfall (and not weather) generators respond your inquiries about practical applications of the proposed approach.
* * *
References:

[revised manuscript text omitted]

---

## Author Response (AR3)

Dear Editor and Reviewers,

Thank you for your new comments about our manuscript entitled "Non-stationary stochastic rain type generation: accounting for climate drivers".

The paper has been revised accordingly. In particular, the supplementary material has been formatted as a technical note, and all figures are now provided with an enhanced resolution.

Hoping that our responses address your concerns, and that our propositions of improvements will fulfil your expectations,

Best regards,

Lionel Benoit, Mathieu Vrac and Gregoire Mariethoz.

[revised manuscript text omitted]
*. It consists of several side experiments that were carried out to support technical details of the main study. More precisely, Supplementary material 1 reiterates the cross-validation of Sect 4.1 in case radar images with a low rain coverage are assigned to rain types, and not classified as dry (cf Sect 2.1). Supplementary material 2 assesses the impact of using high-resolution and in-situ data as meteorological covariates, instead of the reanalysis data introduced in Sect 2.2. Supplementary material 3 explores the use of a parametric model as an alternative to the non-parametric model detailed in Sect 3. Finally, Supplementary material 4 details the bias correction of RCM data prior to their use for precipitation downscaling in Sect 5.

**Supplementary material 1: Cross-validation when accounting for low rain coverage**

In this supplementary material, we reiterate the cross-validation of Sect 4.1 for rain type time series in which radar images with a low rain coverage (rain fraction between 5% and 10%) are regarded as wet. To build such time series, radar images with more than 10% rainy pixels are first classified into rain types according to their space-time-intensity statistical signature as detailed in Sect 2.1. However, in contrast with the main study, the images with rain fraction between 5% and 10% are classified into rain types in a second step by assigning them the type of the closest classified image (i.e. nearest neighbor interpolation in time, cf. (Benoit et al., 2018b)).

The exact same leave-one-year-out cross-validation procedure than in Sect 4.1 is applied to the above rain type time series: for a given simulation year, the rain type model is first trained using data from the 2001-2017 period, excluding the year to simulate. Next, 50 realizations of rain type time series are generated for the year of interest by MPS simulation, conditioned to meteorological covariates. This procedure is iterated for each year of the test period (i.e. 2001-2017), and 50 realizations of 17-years long rain type time series are obtained by concatenating in time the 17 yearly simulations.

Results in Fig. 1 show that classifying radar images with low coverage in a rain type instead of considering them dry does not influence the performance of rain type simulation. Indeed, the only noticeable difference with the Fig 6 of the main manuscript is the correction of the dry bias in the present case. Hence, the fact that the performance of rain type simulation is insensitive to the way images with low rain coverage are classified opens the doors to re-adjusting the wet/dry balance by post-processing, as proposed in Sect 2.1.

[Figure]

**Figure 1.** Results of the cross-validation experiment when images with low rain coverage (rain fraction between 5% and 10%) are regarded as wet. (a) Seasonality of rain (and dry) type occurrence (Seasons are DJF (light blue), MAM (pink), JJA (red) and SON (yellow)), (b) rain type persistence, and (c) probability of transition between rain types. Observations are in red and are obtained by assigning the type of the closest classified image to epochs with rain fraction between 5% and 10%. Simulations are in blue and are obtained by propagating the closest rain type to the beginning and to the end of each rain event. In simulations, continuous lines represent the median of the simulated ensembles (50 realizations), and dashed lines represent the Q10 and Q90 quantiles.

**Supplementary material 2: Selection of meteorological covariates**

The stochastic rain type generator developed in this paper requires meteorological covariates in order to (1) ensure the climatological consistency of the simulations, and (2) reproduce the annual cycle as well as the inter-annual variability of rain type occurrence. As mentioned in Sect. 2.2 we focus on meteorological parameters that are known to influence the triggering

5  and the behavior of rain storms (Vrac et al., 2007; Willems, 2001; Rust et al., 2013), namely: pressure, temperature, relative humidity, and wind direction and intensity. Here we choose to use the actual values of the covariates rather than weather types defined by classification of the spatial patterns of one or several of these covariates (Vrac et al., 2007; Rust et al., 2013; Milrad et al., 2014). There are two main reasons for this: First, in case of weather types derived from several covariates, each weather type combines in an intractable manner the influence of the different meteorological parameters, thus making the identification

10  of the climatological drivers of rain type occurrence difficult. Second, the use of weather types drastically reduces the dimensionality of the covariate space, which allows fewer nuances in the links between meteorological conditions and rain types.

Rain type data and simulations are used at 10-min resolution, and therefore the meteorological covariates must be provided at the same resolution. In addition, rainfall is often related to short time changes in meteorological conditions (in particular temperature and air pressure), and high resolution covariates are therefore expected to better explain rain types. However, one

15  of the two foreseen applications of the proposed stochastic rain type generator is the stochastic downscaling of regional climate model projections, which are most of the time available only at daily resolution. Hence the hybrid solution adopted in this study, which uses daily resolution meteorological covariates and disaggregates them to a 10-min resolution as described in Sect. 2.2.

Figure 2 compares disaggregated data derived from the ERA5 reanalysis with in-situ observations carried out by a weather

20  station (located in Erfurt, within the study area) at high resolution (10-min for pressure, temperature and relative humidity, and 6h for wind). Results show that pressure and temperature have similar behaviors in the two datasets, while relative humidity and wind present significant dissimilarities. Regarding relative humidity, the observed dissimilarities are mostly caused by the influence of the daily temperature cycle in the high resolution dataset. When replacing relative humidity by vapor pressure, which is not correlated with temperature, it appears that the differences between the two datasets are considerably reduced.

25  Hence, the independent information (i.e. not correlated with other covariates) carried by the relative humidity does not drastically differ between the two datasets. Regarding wind-related covariates (here the Eastward and Northward components of the wind vector are assessed), it is interesting to note that the dataset derived from the reanalysis carries more signal than the in-situ observations. This is because in-situ observations refer to low altitude winds, which are much more variable than the wind at 850hPa extracted from the reanalysis. Hence, the signal-to-noise ratio of the weather station dataset is lower than

30  the one derived from ERA5. All in all, the proposed disaggregation method performs well to capture sub-daily variations of meteorological conditions, despite some mismatches during stormy periods.

To ensure that the small fluctuations of the meteorological covariates that are missed by the disaggregated dataset do not negatively affect the simulation of rain types, we performed an additional cross-validation experiment where the rain type simulation is forced by the two datasets of covariates described above, namely: (1) ERA5 data disaggregated at 10 min resolution

(i.e. the dataset used throughout the paper) and (2) in-situ observations from a weather station located within the study area. Fig. 3 summarizes the results of this experiment and shows very little differences between the two simulations. The almost similar performance when using the two sets of covariates can on the one hand be explained by the fact that the additional signal embedded into in-situ observations hardly emerges from the measurement noise, and on the other hand by the fact that the additional information brought by the wind data in the reanalysis dataset compensates for the uncertainties related to the 10-minute reconstruction of pressure, temperature and relative humidity data. Finally, since both sets of covariates lead to similar performances, we favor the disaggregation of daily data because it is compatible with the targeted application of regional climate model downscaling.

After disaggregation, the meteorological covariates can be used to investigate how weather conditions influence rain type occurrence. To this end, Fig. 4 displays the impact of pairs of covariates on rain type occurrence. It shows that although temperature and wind intensity are the main drivers for rain type occurrence, the joint knowledge of all covariates is important to assess rain type distribution.

[Figure]

**Figure 2.** Comparison between in-situ observations (red) and disaggregated ERA5 reanalysis data (black) of the meteorological covariates considered in this study. Period of interest: October 2003. From top to bottom: observed rain types (derived from radar images, cf Sect. 2.1, pressure, temperature, relative humidity, vapor pressure, Eastward wind component, Northward wind component.

[Figure]

**Figure 3.** Cross-validation using the leave-one-year out method described in Sect. 4.1 applied to the 2001-2005 period and two different sets of covariates. Observations are in red, simulations using the disaggregated ERA5 covariates are in dark blue, and simulations using in-situ observations of covariates are in light blue. Continuous lines represent the median of the simulated ensembles (30 realizations), and dashed lines represent the Q10 and Q90 quantiles.

[Figure]

**Figure 4.** Joint probability of rain type occurrence conditional to pairs of meteorological covariates. Rows correspond to different rain types, and columns correspond to different pairs of meteorological covariates. The title of the rows indicates the pair of covariates of interest; the first covariate defines the abscissa axis, and the second the ordinate axis. T denotes temperature, P is pressure, H is relative humidity, $W_i$ is wind intensity and $W_d$ is wind direction. In each graph, the probability of rain type occurrence is coded by the color-scale.

**Supplementary material 3: Markov-chain-based model of rain type occurrence**

As mentioned in Sect. 3, our first attempt of building a stochastic rain type model was based on a parametric approach using a Markov-chain model. Despite not fully satisfactory results, this model is presented hereafter as a possible benchmark for the non-parametric framework proposed in the main paper (Sect. 3).

The parametric approach that has been tested to model rain type occurrence builds on Markov-chain models, which were originally developed to model dry/wet sequences in daily resolution applications (Richardson, 1981; Wilby, 1994; Wilks and Wilby, 1999). The existing models are substantially amended to match with the features of sub-daily resolution rain type time series, which significantly differ from dry/wet sequences at daily resolution. To model sub-daily resolution rain type time series, we adopt a non-homogeneous semi-Markov model (i.e. with non-stationary transition probabilities and time-varying step lengths) with N+2 states (Fig. 5). Among these N+2 states, N states model rain types (in the present case N=6), and 2 states model dry periods that are split into 'short dry' (duration <24h) and 'long dry' (duration >24h) states. In practice, if a dry period exceeds 24h, it is split into as many 'long dry' spells as possible, and the remaining time is distributed into two 'short dry' spells of equal length at the beginning and at the end of the overall dry period. The 'long dry' state can only transition to 'short dry', and the rain types (as well as the 'short dry' type) can transition to each other and with the 'short dry' type. A semi-Markov approach (Foufoula-Georgiou and Lettenmaier, 1987; Bárdossy and Plate, 1991) is used to account for the persistence of rain (and 'short dry') types. In our model, the duration of these types is explicitly defined by a Probability Density Function (PDF) of event duration, and the Markov chain is not allowed to be twice in the same state (i.e. transitions from state i to i are censored). On the contrary, the 'long dry' state always lasts exactly 24h, but is allowed to transition to itself to generate long lasting dry spells. Finally, to account for the non-stationarity of rain type occurrence in time, the semi-Markov chain is made non-homogeneous (Hughes and Guttorp, 1999; Vrac et al., 2007). It consists of changing the probability transition matrix of the Markov chain over time conditionally to a set of meteorological covariates X:

$$P(S_t = j | S_{t-1} = i, X_t) \propto \gamma_{ij} . \exp\left( -\frac{1}{2}(X_t - \mu_{ij})\Sigma^{-1}(X_t - \mu_{ij})^T \right) \qquad (1)$$

Where $S_t$ is the state of the Markov chain at time $t$, $\Sigma$ is the covariance matrix of the covariates, $\mu_{ij}$ is the mean vector of the covariates when the transition from type i to type j occurs, and $\gamma_{ij}$ is the baseline (i.e. long term averaged) transition probability from state i to state j. It should be emphasized here that since the 'long dry' state is allowed to transition to itself, the probability of transition from 'long dry' to 'long dry' is driven by the meteorological covariates, and indirectly the length of dry spell duration is made dependent on the state of the atmosphere. Conversely, the persistence of all other states (i.e. rain types and 'short dry' type) is stationary in time, and only the probabilities of occurrence of these states depend on meteorological conditions. The non-homogeneous semi-Markov model of rain type occurrence is summarized in Fig. 5.

The parameters of the model can be inferred from historical records using the following procedure. First, the baseline transition matrix is estimated by counting all transitions between each pair of rain types (incl. dry states) occurring in the calibration dataset, and by normalizing the result by the total number of transitions. Then, the parameters required to make the transition matrix non-homogeneous (i.e. $\mu_{ij}$ and $\Sigma$, cf Eq. (1)) are estimated using a synchronous record of covariate observations. More

[Figure]

**Figure 5.** Schematic view of the non-homogeneous semi-Markov model used to model sub-daily rain type occurrence.

precisely, $\Sigma$ is the empirical covariance matrix and $\mu_{ij}$ is the empirical mean of the covariates for the time steps where the transition from i to j occurs. Finally, the PDFs of rain type duration are assumed to be gamma distributions whose parameters are inferred by likelihood maximization using the observed rain type durations. After inference of model parameters, the calibrated model can be used to generate synthetic rain type time series conditioned to meteorological covariates. To this end, time

5    series of covariates have to be available for the target simulation period. Next, synthetic rain types are stochastically generated by iteratively (i) simulating a rain type (or dry / wet) transition, and (ii) generating the duration of the current event by sampling the PDF of duration of the rain (or dry) type of interest.

Figure 6 shows the results of the same cross-validation experiment than in Sect. 4.1 of the main study, but applied to the parametric model described above. Results show that this model generates a strong dry bias (ratio simulated/observed rain

10    frequency = 0.61) and does not properly capture the inter-annual variability of rain occurrence (correlation between observed and simulated time series = 0.4). This can be explained by the fact that the relationships between the meteorological covariates and the presence of rain are probably more complex than the linear relationship assumed in the non-homogeneous Markov chain formulation of the parametric model, and by the fact that semi-Markov models do not allow for an easy conditioning to continuous-time covariates. These hypotheses are reinforced by the fact that simulations driven by daily mean temperature only

15    do not generate a dry bias (not shown here). To conclude, it should be noted here that the lower performance of the parametric model is due to the complexity of the problem at hand, rather than a deficiency in the model itself, which extends to high temporal resolution problems some approaches that are state-of-the-art in daily resolution stochastic weather generators.

[Figure]

**Figure 6.** Results of the cross-validation experiment for the parametric model. (a) Seasonality of rain (and dry) type occurrence (Seasons are DJF (light blue), MAM (pink), JJA (red) and SON (yellow)), (b) rain type persistence, and (c) probability of transition between rain types. Observations are in red and simulations in blue. In simulations, continuous lines represent the median of the simulated ensembles (50 realizations), and dashed lines represent the Q10 and Q90 quantiles.

**Supplementary material 4: RCM bias correction and performance of associated rain type simulations**

Meteorological covariates are used to encode rain type non-stationarity (Sect 3 of the main study). When used in the context of a changing climate, these covariates are derived from regional climate model (RCM) outputs. In the present case, the meteorological covariates are derived from the Regional Atmospheric Climate MOdel of the Dutch national weather service (RACMO-KNMI (Van Meijgaard et al., 2008)) driven by the CNRM-CM5 Earth system model (Voldoire et al., 2013) forced according to the RCP8.5 emission scenario. However, like almost all RCM projections, the specific RCM that is used to derive meteorological covariates can be biased. The meteorological covariates must therefore be bias-corrected before further use for stochastic rain type simulation (Sect 5 of the main study). In the present study, we apply the CDF-t method for bias-correction of each variable separately (Vrac et al., 2012).

Figure 7 shows the impact of bias-correction on the distribution of each meteorological covariate. One can notice the significant impact of bias-correction on daily minimum temperature and on humidity. Figure 8 shows that after bias-correction, the performance of the model to simulate rain types in the present climate is almost identical for meteorological covariates derived from the RACMO-KNMI RCM and the ones derived from the ERA-5 reanalyses. This result paves the way to precipitation downscaling through stochastic rain type simulation, as detailed in Sect 5.

[revised manuscript text omitted]